# InstructFlow: Adaptive Symbolic Constraint-Guided Code Generation for Long-Horizon Planning

**Haotian Chi**[1,3]   **Zeyu Feng**[2]   **Yueming Lyu**[2]   **Chengqi Zheng**[2]   **Linbo Luo**[5]
**Yew-Soon Ong**[2,4]   **Ivor Tsang**[2,4]   **Hechang Chen**[1,3†]   **Yi Chang**[1,3†]   **Haiyan Yin**[2†]

[1]School of Artificial Intelligence, Jilin University, China
[2]CFAR and IHPC, Agency for Science, Technology and Research (A*STAR), Singapore
[3]Engineering Research Center of Knowledge-Driven Human-Machine Intelligence,
Ministry of Education, Jilin University, China
[4]Nanyang Technological University (NTU), Singapore, [5]Xidian University, China
chiht21@mails.jlu.edu.cn    {chenhc, yichang}@jlu.edu.cn
{feng_zeyu, lyu_yueming, ivor_tsang, yin_haiyan}@a-star.edu.sg
zhen0144@e.ntu.edu.sg  asysong@ntu.edu.sg  lbluo@xidian.edu.cn,

## Abstract

Long-horizon planning in robotic manipulation requires translating under-specified, symbolic goals into executable control programs that satisfy spatial, temporal, and physical constraints. However, existing language model-based planners often struggle with decomposing long-horizon tasks, enforcing constraints robustly, and adapting effectively to execution failures. We introduce **InstructFlow**, a multi-agent framework that establishes a *symbolic, feedback-driven flow* for code generation in robotic manipulation. InstructFlow comprises three coordinated agents: a **InstructFlow Planner** that constructs and traverses a hierarchical instruction graph to decompose goals into semantically grounded subtasks; a **Code Generator** that synthesizes executable code snippets conditioned on this graph; and a **Constraint Generator** that analyzes execution feedback to induce symbolic constraints when execution failures occur. These constraints are propagated upstream to refine the instruction graph and guide localized code revision without full regeneration. This graph-guided, dynamic flow enables structured, interpretable, and failure-resilient planning, yielding substantial improvements in task success rate and robustness across diverse manipulation benchmarks, particularly in constraint-sensitive and long-horizon scenarios. The implementation is available at https://github.com/chiht21/InstructFlow.

## 1 Introduction

Large language models have emerged as a central paradigm for robotic code generation, translating natural language instructions into executable control programs [1, 14, 19, 27]. Despite their versatility, they struggle with long-horizon task decomposition, reliable constraint enforcement, and adaptive failure recovery. In a manipulation task such as placing an object into a bowl, an LLM-based planner may detect that a grasp fails due to a collision with stacked objects. However, it lacks the capacity to reason beyond the immediate failure and infer its structural cause. Existing approaches often resort to blind retries or ad-hoc replanning, generating similar trajectories that reproduce the same error. These behaviors reveal a critical gap: LLMs can recognize execution failures but lack the structured reasoning required to interpret and repair them.

---

[†]Corresponding authors.

39th Conference on Neural Information Processing Systems (NeurIPS 2025).

This limitation arises from the representational nature of language itself. Natural language instructions are inherently under-specified, ambiguous, and difficult to ground in the physical world. When tasked with generating full execution plans directly, LLMs frequently produce syntactically correct but physically infeasible code, and they rarely recover once execution errors occur. Recent methods have sought to bridge this gap by generating grounded skill sequences with continuous parameters [30] or synthesizing end-to-end executable programs [14]. Yet these approaches remain flat and reactive, lacking the hierarchical structure and feedback integration required for consistent reasoning under dynamic, constraint-rich conditions.

Recent works have begun exploring feedback-driven planning, where language models incorporate environmental signals to improve execution reliability. This is exemplified by PRoC3S [3], which introduces a two-phase pipeline that separates plan generation from constraint checking, enabling failure-triggered replanning when constraint violations occur. While PRoC3S strengthens robustness through feedback, it still treats failures at the surface level, reacting to violations without interpreting their deeper structural causes. This limitation stems from language-based plans being under-specified and unstructured, thereby limiting causal reasoning and systematic plan refinement. Addressing this gap demands a framework that can induce symbolic knowledge from failure and use it to guide targeted, compositional repair rather than wholesale regeneration.

To address these challenges, we introduce **InstructFlow**, a modular, multi-agent framework that establishes a symbolic, feedback-driven information flow for adaptive task planning and code generation. InstructFlow operates beyond surface-level feedback: it interprets execution failures, induces symbolic constraints that capture their causal structure, and propagates this knowledge through a hierarchical instruction graph that organizes task goals into composable subplans. Its constraint induction mechanism abstracts execution traces into human-interpretable predicates that encode spatial, relational, and physical dependencies across tasks. By feeding these symbolic constraints back into the planning hierarchy, InstructFlow performs targeted repair instead of full regeneration, transforming reactive trial-and-error into deliberate, compositional reasoning and achieving interpretable, failure-resilient planning in long-horizon, constraint-sensitive robotic environments.

We highlight three key contributions of this work: (i) a modular multi-agent framework that establishes a symbolic, feedback-driven flow of reasoning for adaptive and interpretable code generation in robotic planning; (ii) a symbolic constraint induction mechanism that abstracts execution failures into causal, reusable predicates, enabling targeted repair rather than full regeneration; and (iii) extensive empirical validation across drawing, block stacking, and YCB packing benchmarks, demonstrating substantial gains in success rate, robustness, and recovery efficiency over strong LLM-based baselines.

## 2 Related Works

**LLM-Based Code Generation for Robotic Manipulation**   Recent advances highlight the potential of LLMs as general-purpose planners for robotic manipulation through code generation. CaP [14] pioneers the use of LLMs to synthesize Python-based reactive controllers, integrating perception modules and control primitives. RoboScript [1] proposes a unified interface for deploying such code across simulation and real robots, focusing on deployability and modularity. LLM$^3$ [30] integrates task and motion planning with LLM-driven failure reasoning for robust code generation in dynamic environments. Instruct2Act [9] and VoxPoser [10] combine LLMs with VLMs, grounding language instructions into actionable code conditioned on perceptual inputs. RoboCodeX [19] introduces a tree-structured multimodal reasoning framework, decomposing language commands into object-centric manipulation code. Recent systems like OctoPack [20] and RobotCode [13] further enhance generalization and reliability by combining LLM-generated programs with skill libraries. While these approaches demonstrate the promise of LLMs when equipped with structured APIs, affordance models, and perceptual grounding, they continue to struggle with handling execution failures and adapting plans in constraint-sensitive or long-horizon tasks.

**Symbolic Abstraction Planning**   Symbolic representations remain crucial for long-horizon and constraint-sensitive manipulation. Classical TAMP systems [11, 2, 4, 7, 26] integrate symbolic task planning with motion controllers but depend on domain-specific predicates. Traditional robotics planning relies on hard-coded symbolic world models [7, 12]. Hybrid methods bridge language and symbolic planning, such as LLM+P [16], which translates instructions into PDDL for optimal symbolic planning. VisualPredicator [15] learns neuro-symbolic predicates from visual inputs, while

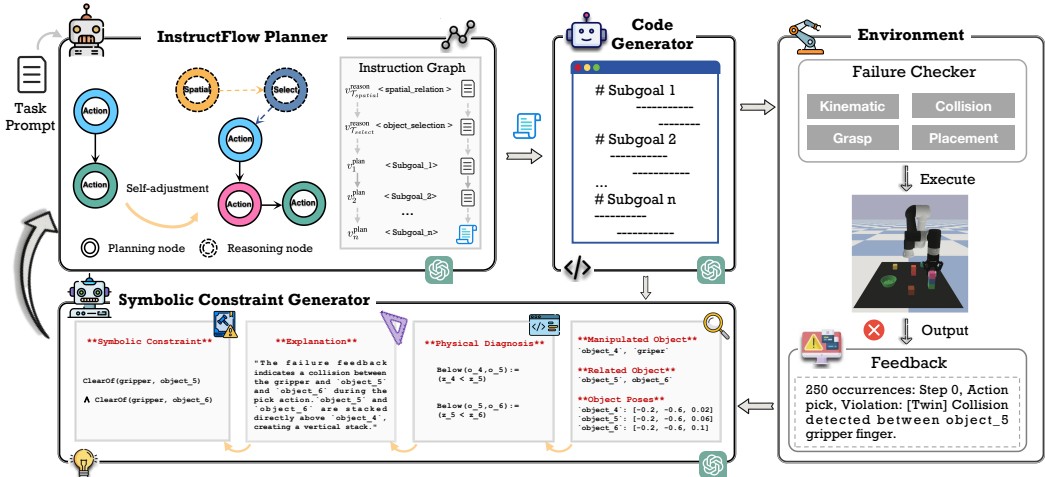

Figure 1: Overview of the **InstructFlow** framework for multi-agent, symbolic, and feedback-driven code generation in robotic manipulation. The system comprises three coordinated agents: (a) **InstructFlow Planner**: parses the task prompt and constructs a multi-level instruction graph that organizes goals into *spatial* and *reasoning* subgoals; (b) **Code Generator**: synthesizes executable code segments and samples parameter domains to instantiate the plan; (c) **Constraint Generator**: analyzes execution feedback and induces symbolic constraints that refine the instruction graph for targeted code correction.

ViLaIn [23] extracts scene-level symbolic representations from vision-language models. Other works like PlanBench [28] create symbolic abstractions for plan feasibility analysis; P3IV [32] and RLang [21] focus on domain-agnostic symbolic representations aligned with LLM reasoning. Despite their use of symbolic abstractions, these approaches lack mechanisms for dynamic symbolic reasoning to abstract task-relevant information or diagnose failures for plan recovery, which is a gap our method explicitly addresses by integrating symbolic reasoning into the code generation and planning loop.

**Feedback-Driven Failure Recovery**    Recovering from execution failures remains a central challenge for LLM-driven robotics. Most approaches treat LLMs or VLMs as success detectors [5, 18, 29], while recent works explore feedback-driven plan repair. REFLECT [17] and AHA [6] leverage LLMs and VLMs for multi-modal failure explanation, enabling language-guided correction. RoboRepair [22] and DoReMi [8] integrate LLMs and VLMs for execution misalignment detection and proactive repair. However, these methods primarily rely on fine-tuning models for failure understanding. To avoid fine-tuning, LLM[3] [30], ProgPrompt [24], and CLAIRify [25] propose failure-aware prompting and runtime verification to guide iterative plan repair. PRoC3S [3] further introduces a hybrid approach that combines LLM-generated partial programs with post-hoc constraint optimization. Building on this foundation, our approach integrates symbolic constraint induction and graph-guided plan repair directly into the code-generation loop, achieving interpretable and adaptive failure recovery that generalizes across tasks.

## 3    Methodology

We begin by formalizing the LLM-based code generation paradigm for robotic manipulation. We then introduce **InstructFlow**, a modular multi-agent framework built around a *hierarchical instruction graph* and a *symbolic constraint induction* mechanism. An overview of the system architecture is provided in Figure 1.

### 3.1    Overview

The problem of code generation for robotic manipulation involves translating natural language instructions into executable programs that operate reliably in robotic environments. Given task descriptions and the initial state of the environment, an LLM is instructed to generate parameterized

action plans that invoke low-level control routines to solve the task. Following a general paradigm [3], the LLM produces two functions per task:

- *get_plan*: a sequence of high-level actions conditioned on free parameters and environment state.

- *get_domain*: the feasible ranges for those parameters (e.g., spatial offsets), defining the search space for plan instantiation.

At the task level, the goal is to generate code that completes the instructed objective without violating physical constraints. Execution feasibility is assessed by a continuous constraint satisfaction program (CCSP) module, which enforces four environment-level checks: *kinematic reachability*, *collision avoidance*, *grasp stability*, and *placement validity*. The LLM must reason not only over a series of *discrete action choices*, but also over *continuous numerical parameters*, which is a nontrivial requirement. Beyond sequencing skills, it must produce long-horizon code grounded in geometry, dynamics, and task-specific semantics. The core challenge lies in the lack of grounding from language prompts to low-level control logic that adheres to physical and dynamic constraints. When execution fails, LLMs often repeat or compound errors, exhibiting limited ability to adaptively repair code.

Our work aims to improve the failure recovery ability of LLM-based planners through structured planning and symbolic constraint-driven code repair.

We propose **InstructFlow**, a modular multi-agent framework for symbolic, feedback-driven code generation in robotic manipulation. The core idea is to introduce an **instruction graph** that hierarchically decomposes high-level task prompts into semantically structured subgoals, and couple this representation with a **symbolic constraint induction mechanism** for effective plan repair. Instruct-Flow modularizes the code-generation pipeline into three cooperative agents, each with a specialized role and local reasoning context (see Appendix A for the prompts used in our three Agents):

- **InstructFlow Planner**: Parses the task prompt and constructs a hierarchical instruction graph capturing semantic and spatial dependencies. Each node encodes a typed subgoal grounded in the robot's skill space.

- **Code Generator**: Translates each subgoal into executable Python code, producing symbolic control routines along with parameter domains for sampling feasible continuous values;

- **Constraint Generator**: Monitors execution failures and induces symbolic constraints that explain the cause. These constraints guide context-specific graph and prompt revisions, enabling targeted subgoal repair without full plan regeneration.

### 3.2 Instruction Graph Construction and Update

**Instruction Graph Semantics** A central architectural contribution of InstructFlow is a hierarchical instruction graph that enables structured task decomposition and adaptive symbolic reasoning. At each interaction round $t$, the **InstructFlow Planner** constructs an instruction graph $\mathcal{G}_t = (\mathcal{V}_t, \mathcal{E}_t)$, conditioned on the task goal, initial state, and symbolic feedback. This graph acts as a typed, declarative scaffold for both task decomposition and constraint-aware refinement. The node set is partitioned as:

$$\mathcal{V}_t = \mathcal{V}_{\text{plan}} \cup \mathcal{V}_{\text{reason}}, \quad \mathcal{V}_{\text{plan}} \cap \mathcal{V}_{\text{reason}} = \emptyset, \tag{1}$$

with edges $\mathcal{E}_t \subseteq \mathcal{V}_t \times \mathcal{V}_t$ capturing symbolic or temporal dependencies. Each edge $(v_i, v_j)$ denotes a directed flow of information, allowing parent nodes to influence the semantic context of their children.

**(1) Planning nodes** $v^{\text{plan}} \in \mathcal{V}_{\text{plan}}$ define grounded subgoals directly translatable into robot-executable code. Each is instantiated as a symbolic prompt:

$$v^{\text{plan}} : (\texttt{goal}, \texttt{state}) \to \texttt{subgoal}_{\mathcal{A}}, \quad \mathcal{A} \in \{\texttt{pick}, \texttt{place}, \ldots\}.$$

These nodes form the plan's executable backbone and anchor structural code generation.

**(2) Reasoning nodes** $v^{\text{reason}}_{\mathcal{T}_j} \in \mathcal{V}_{\text{reason}}$ perform typed symbolic transformations that enrich planning with task-level abstraction and constraint resolution. These typed modules abstract reusable domain knowledge, enabling modular plan revision: $v^{\text{reason}}_{\mathcal{T}_{select}} : \mathcal{I}_{\mathcal{T}_j} \to \mathcal{O}_{\mathcal{T}_j}$, where $\mathcal{I}_{\mathcal{T}_j}, \mathcal{O}_{\mathcal{T}_j}$ denote structured symbolic fields. Outputs are propagated to downstream planning nodes, injecting symbolic knowledge

such as spatial adjacency or parameter tuning. We instantiate five core reasoning modules:

$$
\begin{aligned}
\mathcal{T}_{\text{spatial}} &: S \rightarrow \texttt{Rel}(\text{Objects}, \text{Adjacency}) && \text{(spatial relation inference)} \\
\mathcal{T}_{\text{density}} &: S \rightarrow \texttt{Rel}(\text{Objects}, \text{Density}) && \text{(local clutter estimation)} \\
\mathcal{T}_{\text{select}} &: (G, S, \Phi) \rightarrow \texttt{Select}(\text{Objects}) && \text{(target object selection)} \\
\mathcal{T}_{\text{plan}} &: (G, S, \Phi) \rightarrow \texttt{Order}(\text{Actions}) && \text{(task logic inference)} \\
\mathcal{T}_{\text{param}} &: (G, \Phi) \rightarrow \texttt{Refine}(\text{ParamDomain}) && \text{(parameter range refinement)}
\end{aligned}
$$

Here, $G$ denotes the high-level task goal, $S$ represents the initial state, and $\Phi$ captures symbolic constraints induced from prior failures. These inputs are used by reasoning nodes to extract task-relevant abstractions for plan refinement.

**Feedback-Driven Graph Update**  A core capability of **InstructFlow** is its ability to revise the instruction graph $\mathcal{G}_t$ based on symbolic constraint feedback and failure diagnostics. At initialization ($\texttt{constraint}_0 = \emptyset$), the graph contains only planning nodes. Upon failure (e.g., collisions, instability), the planner inserts reasoning nodes upstream of affected subgoals, dynamically composing a symbolic stack tailored to the error mode:

$$
\mathcal{G}_t = \texttt{Planner}_{\text{LLM}}(\texttt{goal}, \texttt{state}_t, \texttt{constraint}_{t-1}). \tag{2}
$$

This mechanism supports coarse-to-fine symbolic planning by injecting only the reasoning needed to refine or repair the faulty part of the task.

**InstructFlow-Guided Code Generation**  InstructFlow translates symbolic plans into executable code by composing structured prompts along the instruction graph $\mathcal{G}_t$. Each planning node $v_i^{\text{plan}} \in \mathcal{V}_{\text{plan}}$ generates a prompt:

$$
\underbrace{\texttt{instr}^{(t)}}_{\text{Task Prompt}} = \text{Encode}\left(\{v_{\mathcal{T}_j}^{\text{reason}}\}_{j=1}^{|v^{\text{reason}(t)}|}, \ v^{\text{plan}}\right), \qquad \underbrace{\texttt{code}^{(t)}}_{\text{Generated Code}} = \text{LLM}\left(\texttt{instr}^{(t)}\right) \tag{3}
$$

where $\text{Encode}(\cdot)$ integrates the subgoal with symbolic refinements from reasoning nodes, such as spatial relations, parameter ranges, or action dependencies. $|v^{\text{reason}(t)}|$ denotes the number of reasoning nodes providing contextual information to $v^{\text{plan}}$, including types such as spatial reasoning, object selection, and other reasoning nodes as defined above.

This symbolic conditioning guides the LLM to produce context-aware and physically valid code. When failures occur, **InstructFlow selectively updates the relevant subgoals and reasoning nodes impacted by the constraint violations**, avoiding unnecessary recomputation of unrelated parts of the plan. By structuring prompt construction around the symbolic instruction graph, InstructFlow achieves interpretable, constraint-compliant code generation, significantly enhancing sample efficiency and robustness in long-horizon, constraint-sensitive tasks.

**The Role of "Flow"**  While InstructFlow introduces multiple agents and a hierarchical instruction graph, the key distinguishing feature lies in the flow of symbolic information and feedback throughout the entire code generation loop. Unlike static prompting approaches, InstructFlow treats the prompt construction itself as a dynamic, graph-guided flow, where high-level goals, reasoning outputs, and failure-induced constraints are progressively injected into task-specific prompts at each planning node. This flow-centric prompt composition ensures that each code snippet is generated in a context-aware, failure-resilient, and constraint-compliant manner, enabling efficient plan repair without full regeneration. The flow mechanism thus operates at two intertwined levels: *Graph-level symbolic flow*: From reasoning nodes to planning nodes. *Prompt-level information flow*: From task goal, through symbolic reasoning and feedback, into structured, adaptive prompts.

### 3.3  Symbolic Constraint Induction from Execution Failures

LLM-based robotic planners often lack structured mechanisms for failure recovery, relying instead on naïve re-prompts or implicit retries. We introduce a **Constraint Generator** that diagnoses execution failures, and abstracts them into logical constraints. These constraints serve as *symbolic corrections that guide graph restructuring and prompt refinement*, enabling interpretable, efficient, and generalizable plan repair.

**Failure Diagnosis Workflow.** The symbolic constraint induction follows a four-stage reasoning workflow: (i) **Failure-relevant entities retrieval**, which identifies failure-relevant entities from the failure trace $\mathcal{F}_t$ and executed code $\mathcal{P}_t$; (ii) **Code-level reasoning**, which instantiates involved variables and reasoning symbolic predicates that reflect the physical feasibility; (iii) **Diagnostic reasoning**, which compute geometric or geometric diagnostics based on predicates, such as collision proximity, path clearance, and placement stability; and (iv) **Symbolic constraint induction**, which abstracts diagnostic findings into declarative symbolic constraints that encapsulate the feasibility conditions violated by the current plan. This structured workflow transforms grounded execution failures into symbolic rules that guide prompt regeneration and enable interpretable, plan repair.

**Physical Predicate as an Induction Basis.** To enable interpretable failure diagnosis and structured symbolic constraint induction, we ground physical feasibility reasoning on a set of declarative physical predicates. These predicates abstract task-specific physical interactions into reusable logical representations, serving as the foundation for symbolic reasoning across diverse manipulation scenarios. We categorize predicates along four functional components:

(i) **Entities** ($\mathcal{E}$): Rather than pre-defining entities rigidly, Instruct-Flow dynamically abstracts task-relevant entities into functional roles based on the evolving task context and feedback, such as ?target (manipulated object), ?neighbor (potential obstacles), ?surface (supporting structures), and ?gripper (robot end-effector); (ii) **Relations** ($\mathcal{R}$): Symbolic relations are flexibly instantiated to capture emergent spatial and semantic interactions during task execution and diagnosis, such as On(?a, ?b) for support/contact, or ClearOf(?a, ?b) for proximity constraints, enabling contextual adaptation rather than relying on static domain rules; (iii) **Physical Functions** ($\mathcal{F}$): InstructFlow leverages a set of physical diagnostics as interpretable abstractions, such as Dist(?a, ?b), SupportArea(?obj), and COMDeviation(?obj), which are dynamically evaluated in response to execution feedback, guiding the symbolic reasoning process without hard-coded thresholds; (iv) **Thresholds** ($\mathcal{B}$): Task-specific feasibility bounds, such as $\delta_{\text{safe}}$ for

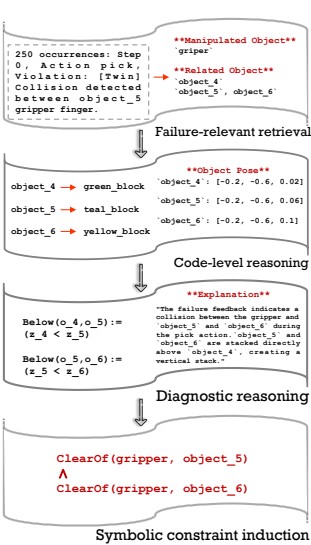

Figure 2: Failure Diagnosis Workflow of Symbolic constraint generator.

clearance margins, and $\eta_{\text{min}}$ for support stability ratios, , which can be tuned or inferred based on the environment state and failure modes, allowing InstructFlow to generalize beyond fixed rule specifications.

These symbolic forms are grounded by diagnostics over physical basis, but are interpreted and manipulated as logical components of the instruction graph, which isolate the physical root causes of failure and ground them in explicit task parameters, forming the basis for symbolic abstraction.

**Symbolic Constraint Induction.** We formalize the symbolic constraint $\phi$ as a conjunction over two complementary modalities of failure correction: relational structure and physical feasibility:

$$\phi := \bigwedge_{c \in \mathcal{C}(\mathcal{E}, \mathcal{R}, \mathcal{F}, \mathcal{B})} c, \quad \text{where} \quad \mathcal{C}(\mathcal{E}, \mathcal{R}, \mathcal{F}, \mathcal{B}) = \underbrace{\{R_i(e_{a_i}, e_{b_i})\}}_{\text{Relational Constraints}} \cup \underbrace{\{f_j(\Theta_j) \oplus \tau_j\}}_{\text{Physical Constraints}}. \quad (4)$$

Here, for each relational constraint $R_i(e_{a_i}, e_{b_i})$, $e_{a_i}$ and $e_{b_i}$ are *entity instances* (e.g., block, bowl) participating in the relation $R_i$. For physical constraints, each term $f_j(\Theta_j) \oplus \tau_j$ represents a *feasibility condition*, where: $\Theta_j$ denotes the *variables* involved (e.g., poses, offsets), $\oplus$ is a *comparison operator* (e.g., $\leq$, $\geq$, or $=$), $\tau_j \in \mathcal{B}$ is a *task-specific threshold* (e.g., maximum allowable clearance).

This formulation allows each constraint $\phi$ to capture both high-level task semantics and low-level physical requirements within a unified logical form, which the **Constraint Generator** can compose into logical constraints $\phi$ for plan repair. For instance:

$$\phi_{\text{pick}} := \text{ProximitySafe}(?object, ?neighbor) \wedge \text{PathClear}(?gripper, ?object),$$
$$\phi_{\text{place}} := \text{Dist}(?pose, ?neighbor) \geq \delta_{\text{safe}} \wedge \text{StableOn}(?object, ?surface).$$

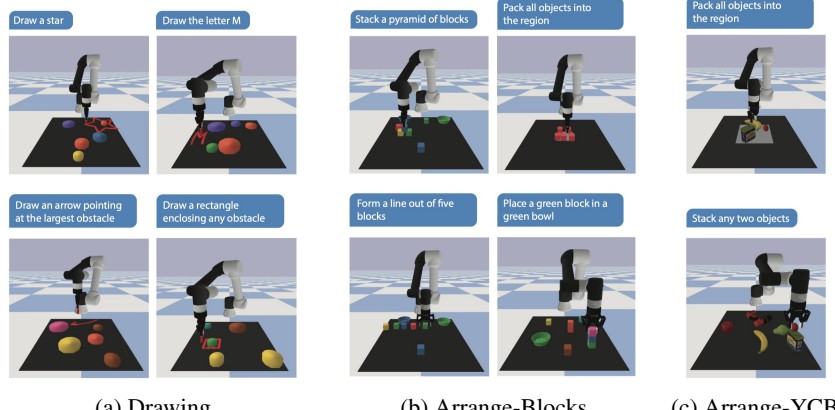

|                   |                   |                   |
| (a) Drawing       | (b) Arrange-Blocks | (c) Arrange-YCB   |

Figure 3: Illustration of tasks in our simulated environments, along with corresponding goals.

Notably, these symbolic constraints act as structured priors for graph refinement and generalize across task instances and environments, enabling not just plan repair but modular, interpretable priors that can be reused across planning episodes. By treating failure correction as symbolic program refinement, this representation integrates seamlessly into our instruction graph and enables feedback-driven, structurally grounded prompt generation (see Appendix B.2.2 for the symbolic constraints we summarized from the experimental results).

## 4 Experiments

### 4.1 Experimental Setup

We adopt the same environments, evaluation metrics, and protocol as PRoC3S [3] to ensure fair comparison, while extending its core planning pipeline with symbolic reasoning and constraint-guided repair. All experiments are conducted in the Ravens [31] simulation environment, using a 6-DoF UR5 arm with a Robotiq 2F-85 gripper in a tabletop workspace. Physics-based execution and constraint checking are handled via PyBullet. Simulations run on CPUs with 32GB RAM, with all baseline implementations integrated into a unified evaluation framework.

**Domains and Tasks.** We evaluate our approach on three simulated domains, each designed to test different aspects of long-horizon planning with parameterized skills and physical constraints:

(1) **Drawing**: The robot is equipped with a `draw_line` primitive that generates 2D trajectories to render geometric and symbolic shapes on a surface, while avoiding randomly placed objects. These tasks require precise parameter coordination under tight spatial constraints.

(2) **Arrange-Blocks**: The robot stacks and arranges colored blocks and bowls to form pyramids, lines, or centered clusters. This domain tests stability, spatial accuracy, and planning under clutter and occlusions.

(3) **Arrange-YCB**: The robot manipulates complex objects from the YCB dataset (e.g., banana, meat can) to perform packing and stacking. Irregular geometries introduce challenges in grasping, placement feasibility, and collision avoidance.

**Constraints.** Across all domains, generated plans are evaluated against a set of physical and geometric constraints that reflect real-world robotic limitations: (1) **Kinematic constraints** ensure that the robot's inverse kinematics solver produces a reachable end-effector pose, rejecting infeasible motions; (2) **Collision constraints** eliminate plans that lead to unintended contact between the robot, environment, or other objects, allowing only expected contact such as during grasps; (3) **Grasp constraints** verify that the gripper properly encloses the object and maintains stability during lifting, rejecting grasps that cause slippage or collision; (4) **Placement constraints** require that, upon release, the object remains upright and stationary, i.e., any post-placement drift or instability signals a failure of physical feasibility.

**Baselines.** We compare our approach against three baselines:

(1) **PRoC3S** [3]: The original two-phase LLM-based planner that separates plan generation and constraint satisfaction using a sampling-based solver with feedback.

(2) **LLM³** [30]: A recent method in which the LLM directly outputs grounded skill sequences with continuous parameters.

(3) **Code-as-Policies (CaP)** [14]: A program synthesis-based strategy that uses an LLM to produce complete Python programs encoding the action sequence and continuous parameters for execution.

**Execution Details.** Each approach is evaluated over 10 randomized seeds per simulated task. We use a maximum budget of 1000 samples per trial (10000 for drawing tasks). We limit the number of feedback iterations to 5. All methods are queried via OpenAI's GPT-4o unless otherwise stated. A task is considered successful if the final robot state satisfies the goal condition without violating any constraints (see Appendix B.1 for more details on experiment settings).

## 4.2 Benchmark Experiments

We benchmark InstructFlow against PRoC3S, LLM³, and CaP across three domains. As shown in Table 1, across drawing, block arrangement, and YCB manipulation tasks, InstructFlow outperforms prior methods by 20–40% in task success rate.

| | Drawing | | | | Arrange Blocks | | | | Arrange YCB | |
|---|---|---|---|---|---|---|---|---|---|---|
| | Star | Arrow | Letters | Enclosed | Pyramid | Line | Packing | Unstack | Packing | Stacking |
| LLM³ | 40% | 40% | 80% | 50% | 0% | 40% | 30% | 0% | 0% | 10% |
| CaP | 10% | 0% | 40% | 30% | 20% | 20% | 20% | 10% | 30% | 10% |
| PRoC3S | 90% | **80%** | 80% | 90% | 60% | 70% | 50% | 60% | 30% | 40% |
| InstructFlow (Ours) | **100%** | **80%** | **100%** | **100%** | **90%** | **100%** | **90%** | **90%** | **60%** | **70%** |

Table 1: Task success rates (%) across drawing, block arrangement, and YCB manipulation domains. Bold indicates top-performing results.

This improvement arises from InstructFlow's ability to perform structured symbolic reasoning over task-specific failures, enabling targeted plan corrections at multiple levels: (i) refining parameter domains to satisfy geometric constraints, (ii) inducing symbolic relations (e.g., adjacency, clearance) to prevent repeated failure modes, and (iii) revising subgoal sequences based on environment feedback. For instance, in the *Pyramid* and *Line* tasks, baseline methods (e.g., PRoC3S) frequently fail due to improper block spacing, leading to unstable stacks. InstructFlow detects these failures and augments the instruction graph with symbolic adjacency constraints (e.g., `Adjacent(?block_i, ?block_j)`), which guide the adjustment of offset ranges and enforce tighter placements, improving stability without exhaustive re-planning.

Similarly, in *Packing* tasks involving YCB objects with irregular geometries, InstructFlow leverages symbolic constraints over object proximity to guide precise placement corrections. When initial plans result in collision-prone configurations, the system identifies violated `ClearOf` constraints and refines placement parameter ranges to balance object clearances and workspace boundaries. This targeted adjustment enables feasible, collision-free placements without exhaustive resampling, a capability that flat prompt-based methods notably lack due to their absence of structured, feedback-driven repair mechanisms. (See Appendix B.2.1 for more experiment results on VLM)

| | Drawing | | | | Arrange Blocks | | | | Arrange YCB | |
|---|---|---|---|---|---|---|---|---|---|---|
| | Star | Arrow | Letters | Enclosed | Pyramid | Line | Packing | Unstack | Packing | Stacking |
| InstructFlow | **100%** | **80%** | **100%** | **100%** | **90%** | **100%** | **90%** | **90%** | **60%** | **70%** |
| InstructFlow w/o Planner Agent | 90% | 80% | 80% | 100% | 50% | 90% | 50% | 40% | 40% | 40% |
| InstructFlow w/o Constraint Agent | 100% | 80% | 100% | 80% | 40% | 100% | 60% | 60% | 30% | 40% |

Table 2: Ablation study results (% task success) highlighting the contributions of the InstructFlow Planner and Symbolic Constraint Generator.

Ablation results in Table 2 highlight the distinct roles of symbolic planning and constraint induction in InstructFlow's performance. Without the Planner, the system loses its ability to structure tasks

hierarchically, resulting in brittle plans and severe failures in multi-step spatial tasks (e.g., *Pyramid*, *Packing*, with up to 50% drops). Removing Constraint Induction disables feedback-driven repair, forcing the model into blind retries that struggle with physical feasibility, leading to 30–40% degradation in cluttered and precision-sensitive tasks. These results confirm that InstructFlow's robustness stems from the synergy of symbolic task decomposition and failure-informed constraint refinement.

## 4.3 Robustness to Real-World Uncertainties

While our study is based on simulation, we explicitly model two key real-world uncertainties, sensor noise and feedback ambiguity, to assess the robustness of InstructFlow under imperfect information, a common challenge in physical robotic systems.

**Perceptual Noise.** To simulate sensor inaccuracies, we inject zero-mean Gaussian noise into object poses. As shown in Table 3, InstructFlow maintains high performance even under severe noise ($\sigma = 0.02$, roughly 50% of object size). For instance, success rates remain above 70% across most tasks, demonstrating strong perceptual robustness.

|  | Pyramid | Line | Packing | Unstack | YCB-Packing | YCB-Stacking | Avg. Drop |
|---|---|---|---|---|---|---|---|
| No noise | 90% | 100% | 90% | 90% | 60% | 70% | - |
| $\sigma = 0.005$ | 90% | 100% | 90% | 90% | 60% | 70% | 0% |
| $\sigma = 0.01$ | 80% | 100% | 80% | 80% | 50% | 60% | 8.3% |
| $\sigma = 0.02$ | 70% | 90% | 70% | 70% | 40% | 50% | 18.3% |

Table 3: Performance under varying levels of perceptual noise.

**Feedback-Layer Noise.** We further evaluate the system's resilience to feedback ambiguity by simulating imperfect feedback, including (1) incorrect object references and (2) incomplete traces with missing object IDs or causal descriptions. As summarized in Table 4, InstructFlow sustains near-baseline performance across tasks, despite corrupted or partial feedback, with average success rate drops limited to 11–16%.

|  | Pyramid | Line | Packing | Unstack | YCB-Pack | YCB-Stack | Avg. Drop |
|---|---|---|---|---|---|---|---|
| Ideal Feedback | 90% | 100% | 90% | 90% | 60% | 70% | - |
| Incorrect Feedback | 60% | 80% | 70% | 60% | 30% | 40% | 16% |
| Incomplete Feedback | 70% | 90% | 70% | 70% | 40% | 50% | 11% |

Table 4: Performance under incorrect and incomplete feedback traces.

These results demonstrate that InstructFlow remains effective without perfect state estimation, key characteristics for real-world deployment. While real-robot experiments are an important direction for future work, these robustness evaluations offer strong empirical evidence of the system's readiness for real-world uncertainty.

## 4.4 Case Study

We take the **Unstack** task as a case study to illustrate the effectiveness of InstructFlow. The goal is to **place the green block into the green bowl**, but the task poses hidden challenges: the green block is often buried beneath a stack, making direct access infeasible. Naive pick attempts cause collisions with blocks above, violating the constraints and leading to failure.

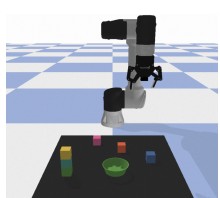

Existing methods, such as PRoC3S, can detect execution failures and make localized repairs, like inserting obstacle removal steps. However, they struggle with multi-layered occlusions. When the green block is buried under multiple stacked objects, PRoC3S lacks a structured mechanism to reason about the correct removal order. As a result, it often generates plans with invalid sequences or actions that reintroduce collisions, ultimately failing to complete the task.

Figure 4: Illustrative image of the environment for the `Unstack`.

**Symbolic Constraints** In the first round of code generation, InstructFlow receives the goal: place the green block into the green bowl. It constructs an initial instruction graph with two planning nodes:

pick `object_4` and place `object_7`. Correspondingly, the code attempts to pick `object_4` at its current pose and place it at the target location.

**Execution feedback**, however, reveals repeated collisions during the `pick` action:

> **[Error Message]**: "250 occurences: Step 0, Action pick, Violation: [Twin] Collision detected between object_5 object gripper finger" and "250 occurences: Step 0, Action pick, Violation: [Twin] Collision detected between object_6 object gripper finger"

Given the **failure feedback** and **generated code**, the Constraint Generator localizes the root cause to the `pick` action on `object_4`, identifying the involved variables: the *manipulated object*, *its pose*, and *the interfering objects* (`object_5`, `object_6`). By analyzing the spatial configuration, InstructFlow infers that the gripper's approach vector intersects with the stacked obstacles, violating collision constraints. This reasoning leads to the generation of explicit **symbolic constraints**:

$$\phi_{\text{unstack}} := \texttt{ClearOf}(\text{gripper}, \text{object\_5}) \wedge \texttt{ClearOf}(\text{gripper}, \text{object\_6})$$

These constraints distill raw collision feedback into symbolic predicates that express the essential condition: the gripper must reach the target without obstruction. This abstraction transforms a low-level failure into a reusable, structured signal for graph updates and targeted code repair.

**Dynamic InstructFlow Graph Update.** Given the symbolic constraints, InstructFlow Planner dynamically updates the instruction graph with reasoning nodes, which model the spatial and logical dependencies in the scene.

Specifically, the updated graph includes a `spatial reasoning node` $v_{\mathcal{T}spatial}^{\text{reason}}$ that analyzes vertical stacking relations between objects, inferring that `object_5` and `object_6` are stacked above `object_4` via symbolic `below` relations. Based on this output, an `object selection reasoning node` $v_{\mathcal{T}selection}^{\text{reason}}$

```
1  def gen_plan(initial: RavenBelief,
2 +              x_clear1: float, y_clear1: float,
3 +              x_clear2: float, y_clear2: float,
4 +              x_offset: float, y_offset: float):
5      plan = []
6
7      # subgoal 1: Pick object_6 (the yellow block)
8      block6 = initial.objects["object_6"]
9      plan.append(Action("pick", block6.pose.point))
10
11     # subgoal 2: Place object_6 at a clear location
12     plan.append(Action("place", [x_clear1, y_clear1, block6.pose.point[2]]))
13 +
14     # subgoal 3: Pick object_5 (the teal block)
15     block5 = initial.objects["object_5"]
16     plan.append(Action("pick", block5.pose.point))
17
18     # subgoal 4: Place object_5 at a clear location
19     plan.append(Action("place", [x_clear2, y_clear2, block5.pose.point[2]]))
20
21     # subgoal 5: Pick object_4 (the green block)
22     block4 = initial.objects["object_4"]
23     plan.append(Action("pick", block4.pose.point))
24
25     # subgoal 6: Place object_4 into object_7 (the green bowl)
26     bowl17 = initial.objects["object_7"]
27     x, y, z = bowl17.pose.point
28     plan.append(Action("place", [x + x_offset, y + y_offset, z]))
29
30     return plan
```

Figure 5: A code snippet illustrating how InstructFlow repairs the `Unstack` plan by intuitively injecting a targeted object removal routine automatically derived from InstructFlow's structural reasoning.

identifies `object_5` and `object_6` as obstacles to be removed according to the induced `ClearOf` constraints. A `logic reasoning node` $v_{\mathcal{T}logic}^{\text{reason}}$ then determines the action sequence that satisfies these constraints, ensuring the objects are unstacked top-down. These reasoning nodes collectively refine the instruction graph by introducing new planning nodes to first move `object_6` (yellow), then `object_5` (teal), and finally pick `object_4` (green), reflecting the inferred symbolic dependencies.

**Code Repair.** Based on the updated instruction graph, Code Generator regenerates the executable code to satisfy the induced symbolic constraints. Unlike black-box retries, the code repair process is explicitly guided by InstructFlow's graph structure, ensuring that prerequisite actions (e.g., obstacle removal) are correctly sequenced before the primary task. As shown in Fig. 5, the repaired code respects both spatial constraints (via clear placement of obstacles) and temporal dependencies (via correct unstacking order), demonstrating InstructFlow's ability to produce interpretable, constraint-compliant programmatic policies. We provide more case studies across different tasks in Appendix B.2.3.

## 5 Conclusions

We presented InstructFlow, a symbolic and feedback-driven framework for robotic code generation that introduces an instruction graph to decompose tasks and enable interpretable, constraint-aware planning. By integrating structured symbolic reasoning and a reusable constraint vocabulary, InstructFlow supports targeted plan repair, avoids full-plan regeneration, and significantly enhances robustness in manipulation tasks. Empirical results across challenging benchmarks validate the system's ability to handle long-horizon, constraint-sensitive scenarios with improved success rates and sample efficiency. Looking ahead, we plan to extend InstructFlow to incorporate visual grounding and multi-modal constraint induction, enabling even richer symbolic reasoning from unstructured feedback in the physical world.

## Acknowledgments

This research is supported by the National Research Foundation, Singapore and Infocomm Media Development Authority under its Trust Tech Funding Initiative, Career Development Fund (CDF) of the Agency for Science, Technology and Research (A*STAR) (No: C233312007, No: C243512014), and the National Research Foundation, Singapore under its AI Singapore Programme (AISG Award No: AISG-NMLP-2024-003), and, in part by the National Natural Science Foundation of China (No. U2341229, No. 62476110); the National Key R&D Program of China (No. 2023YFF0905400, No. 2021ZD0112500); the Key R&D Project of Jilin Province (No. 20240304200SF); the Key R&D Program of Shanxi Province, China (2025GH-YBXM-020). Any opinions, findings and conclusions or recommendations expressed in this material are those of the authors and do not reflect the views of the National Research Foundation, Singapore, and Infocomm Media Development Authority.

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

# Appendix

This appendix provides detailed information about our method and experimental setup. It is organized as follows:

- In section A, we first describe the prompting scheme used by different agents in InstructFlow, including both the shared initialization prompt and agent-specific templates for the planner, code generator, and symbolic constraint generator.

- In section B.1, we provide additional details on our experimental setup, including how programs are instantiated and executed within the simulation pipeline. We also identify critical limitations in the original PRoC3S evaluation protocol and introduce a VLM-based semantic check to address them.

- In section B.2.1 , we present an extended experiment exploring the effect of visual inputs on planning, followed by a summary of symbolic constraints induced across different manipulation tasks in section B.2.2.

- In section B.2.3, we include several representative case studies that illustrate how InstructFlow recovers from planning failures through symbolic reasoning and instruction graph-guided code repair.

# A InstructFlow Prompting Details

Here we provide details on the prompting scheme used for each agent in InstructFlow. The prompting template consists of two parts: a shared initialization template and agent-specific prompts. The complete structure is illustrated as follows:

```
Shared Prompt Templates:
```

```
1 {{{system_prompt}}}
2 {{{domain_setup_code}}}
3 {{{skill_preface}}}
4 {{{domain_skills}}}
```

```
Agent-Specific Prompt Templates:
```

```
1 {{{planner_role}}}
2 {{{code_generator_role}}}
3 {{{constraint_generator_role}}}
```

## A.1 Shared Prompt Templates

All agents share a common initial prompt structure, consisting of: (a) `system_prompt`, (b) `domain_setup_code`, (c) `skill_preface`, (d) `domain_skills`. This shared context is constructed following the initial prompt setup introduced in PRoC3S [3], where detailed environment and skill specifications are defined. We adopt the same structure without modification, and refer readers to the original paper for full specification details.

## A.2 Agent-Specific Prompt Templates

As outlined in Section 3, we have three primary agents: the **InstructFlow Planner**, which decomposes the high-level goal into structured subgoals through instruction graph construction; the **Code Generator**, which translates the instruct graph into executable code snippets; and the **Constraint Generator**, which analyzes execution feedback to induce symbolic constraints for graph-guided code repair.

### A.2.1 InstructFlow Planner

---
**Instruction Graph Construction Prompt**

```
<inputs>
    {task_goal}
    {initial_state}
</inputs>
```

You are responsible for constructing an adaptive instruction graph that serves as an intermediate reasoning structure for robotic task planning. Your task is to generate a sequence of base planning nodes that decompose the task into subgoals. These nodes form the initial executable backbone of the plan. Do not include any reasoning nodes at this stage. Each node should include a semantic description of the subgoal, which can represent either:

- a concrete manipulation action (e.g., pick, place), or
- a prerequisite operation (e.g., selecting a target object, removing an obstacle).

Ensure subgoals are:
- logically ordered,
- collectively sufficient to achieve the goal.

*# Node Type: Planning Node*
{ "id": "n1",
 "type": "action",
 "inputs": ["goal", "initial_state"],
 "output": <"natural language subgoal">}

---

## Instruction Graph Revision Prompt

```
<inputs>
   {task_goal}
   {initial_state}
   {last_instruction_graph}
   {symbolic_constraint}
</inputs>
```

Your task is to revise the instruction graph in response to symbolic constraints inferred from execution failures.This process involves constructing a reasoning flow that incrementally generates a refined instruction to update downstream planning nodes. You need to follow these steps:

Step 1: Reasoning Node Selection
You need to insert one or more reasoning nodes upstream of affected planning nodes guiding by symbolic constraints. Each reasoning node should perform a symbolic transformation relevant to the failure, and output an intermediate instruction fragment.

Step 2: Instruction Flow Construction
Sequentially process each reasoning node, using its output as an instruction fragment that incrementally updates the evolving task description. These fragments form a directed information flow. At the end of the reasoning node, concatenate all instruction fragments to form a single composite instruction that encodes the full reasoning chain and can be used to update downstream planning nodes.

Step 3: Instruction Graph Update
Use the composite instruction to update the planning nodes.  Replace the original subgoal with this revised version, which incorporates both the task intent and refinements.  The updated subgoals of instruction graph should then be passed as the final instruction to the code generator.

We define five types of reasoning nodes, each with its specific functionality and structure as described below:

-The spatial relation reasoning node analyzes pairwise spatial relationships between all visible objects to generate a spatial relation graph:
 *# Node Type: Spatial Relation Reasoning Node*

{ "id": "n2",
 "type": "spatial_relation_reasoning_node",
 "input": ["initial_state"],
 "output": <"spatial_relation_graph">}

-The object density reasoning node estimate the local spatial density around each object to reflect how crowded its surroundings are.
 *# Node Type: Object Density Reasoning Node*

{ "id": "n3",
 "type": "object_density_analysis_node",
 "input": ["initial_state"],
 "output": <"object_density_map">}

-The object selection reasoning node combines the goal, the output of the spatial perception node, the output of the object density analysis node as the inputs to infer which correct manipulated object(s) should be selected to accomplish the goal.
 *# Node Type: Object Selection Reasoning Node*

{ "id": "n4",
 "type": "object_selection_reasoning_node",
 "input": ["goal", "spatial_relations_graph", "object_density_map", "symbolic_constraint"],
 "output": <"manipulated_objects">}

-The plan logic Reasoning Node combines the goal, the output of object selection reasoning node, and the symbolic predicate as the inputs to infer the correct execution order among the manipulated targets.
 *# Node Type: Plan Logic Reasoning Node*

{ "id": "n5",
 "type": "plan_logic_reasoning_node",
 "input": ["goal", "manipulated_objects", "symbolic_constraint"],
 "output": <"execution_order">}

-The parameter range reasoning node combines the goal and the symbolic predicate to adjusts the ranges of action plan parameters based on explicit rules defined in symbolic predicates, and outputs instructions indicating whether to expand or shrink the ranges to meet task requirements.
 *# Node Type: Parameter Range Reasoning Node*

{ "id": "n6",
 "type": "parameter_range_adjustment_node",
 "inputs": ["goal", "symbolic_constraint"],
 "output": <"range_adjustment_instruction">}

### A.2.2 Code Generator

```
<inputs>
   {task_goal}
   {initial_state}
   {subgoals_instruction}
</inputs>
```

You are a code generation agent in a robotic planning system. Your goal is to generate two things:

First, generate a python function named 'gen_plan' that can take any discrete or continuous inputs. No list inputs are allowed and return the entire plan with all steps included where the parameters to the plan depend on the inputs. The plan should be generated based on the initial high-level computation graph, which is composed of a sequence of subgoals. Each subgoal corresponds to either a manipulation action or a prerequisite operation.

Second, generate a python function 'gen_domain' that returns a set of bounds for the continuous or discrete input parameters. The number of bounds in the generated domain should exactly match the number of inputs to the function excluding the state input.

The function you give should always achieve the goal regardless of what parameters from the domain are passed as input. The 'gen_plan' function therefore defines a family of solutions to the problem. Explain why the function will always satisfy the goal regardless of the input parameters. Make sure your function inputs allow for as much variability in output plan as possible while still achieving the goal. Your function should be as general as possible such that any correct answer corresponds to some input parameters to the function.

The main function should be named EXACTLY 'gen_plan' and the domain of the main function should be named EXACTLY 'gen_domain'. Do not change the names. Do not create any additional classes or overwrite any existing ones. Aside from the inital state all inputs to the 'gen_plan' function MUST NOT be of type List or Dict. List and Dict inputs to 'gen_plan' are not allowed. Additionally, the input to 'gen_domain' must be exactly the 'initial:RavenBelief' argument, even if this isn't explicitly used within the function!

### A.2.3 Symbolic Constraint Generator

```
<inputs>
   {task_goal}
   {initial_state}
   {failure_feedback}
   {generated_code}
</inputs>
```

You are a symbolic reasoning agent tasked with diagnosing execution failures in robotic manipulation tasks. Your goal is to induce generalizable symbolic constraints that explain the failure and can guide future plan correction. You task is to perform the following reasoning steps to generate symbolic constraint(s) for plan repair:

Step 1. Failure-relevant retrieval:
Identify the exact code segment responsible for triggering the failure. In a failure feedback, "Step N" refers to the N-th action generated by the code (zero-based index).Specifically, it corresponds to the N-th plan.append(Action(...)) call in the code. For example: "Step 1, Action: place" refers to the second action in the plan (index = 1).

Step 2. Code-level reasoning:
Extract all variables relevant to this failure, including:

-Manipulated objects.
-Object poses and spatial relations.
-Action parameters (e.g., offsets).
-Domain ranges for parameters.

Step 3. Diagnostic reasoning:
Based on the environment state and extracted variables, analyze the geometric or physical cause of failure. You must consider multiple possible causes, including but not limited to:

- Geometric violations (e.g., collisions, unstable placement, path obstruction).
- Temporal inconsistencies (e.g., incorrect subgoal ordering, premature actions).
- Symbolic logical errors (e.g., wrong object selection, missing prerequisite conditions).

For physical causes, compute diagnostic metrics, including but not limited to:
- Proximity distances between relevant entities.
- Existence of collision-free paths.
- Stability metrics (e.g., center of mass projection).

For symbolic or logical causes, analyze:
- Whether the current action respects task-specific symbolic constraints.

- Dependencies between subgoals as defined in the instruction graph.
- Whether preconditions for the current subgoal are satisfied.

Provide a concise diagnosis explaining why the failure occurred, explicitly stating:
- The type of cause (physical, temporal, symbolic).
- The reasoning process leading to this conclusion.

You are not allowed to assume the cause based solely on the failure description. All conclusions must be verified through concrete reasoning over environment state,task semantics, or diagnostic metrics.

Step 4. Symbolic Constraint Induction:
Given the diagnosis and variables, formulate a symbolic constraint that abstracts the failure into a general, reusable rule. You must express the constraint using standard symbolic predicates, including but not limited to:

-Spatial relations: On, ClearOf, Aligned, StableOn...
-Temporal/ordering: Before, Precondition, Order...
-Semantic affordances: Reachable, Occludes, Graspable, Affords...
-Physical feasibility: ProximitySafe, PathClear, PlacementFeasible, ForceStable...

The constraint must express logical conditions involving task entities, their relationships, or parameter ranges, and be parameterized with thresholds or bounds when applicable.

# B  Experiment Details and Additional Results

## B.1  Experimental Setup Details

### B.1.1  Program Instantiation and Simulation Pipeline

Our framework follows the same two-stage code generation and execution process used in PRoC3S [3]. Specifically, after generating the code that defines a task plan via LLM, the resulting Python function consists of two components: `get_plan()` and `get_domain()`.

The `get_plan()` function encodes a sequence of symbolic actions with continuous or discrete parameters, corresponding to the subgoals decomposed in the instruction graph. The `get_domain()` function specifies the sampling bounds for each parameter from a predefined sampler. These samplers (e.g., `ContinuousSampler`, `GraspSampler`) generate candidate values for plan parameters without awareness of environmental constraints such as collisions or instability.

To ground the abstract plan into an executable one, we adopt the same strategy as PRoC3S: sample $n$ parameter instantiations from the domains defined in `get_domain()`, and for each instantiation, evaluate the resulting plan in a physics-based simulator. If the plan violates any constraints (e.g., Kinematic, collisions, grasp, placement constraints), the simulator reports detailed constraint violation feedback. Once a constraint-free plan is found in simulation, it is deployed in the real environment.

### B.1.2  Fixing Protocol Limitations in PRoC3S Evaluation

While our experimental setup builds directly on the original PRoC3S [3] framework and reuses its environment, skill library, and simulation interface, we identified structural limitations in its evaluation protocol. Specifically, PRoC3S treats any execution that does not explicitly violate simulator constraints as successful, regardless of whether the task goal has been semantically achieved.

**Incorrect but constraint-free Plan Misclassified as Successful**  For example, a plan may place an object in an incorrect position, fail to form the required structure (e.g., a pyramid), or manipulate the wrong object altogether. As long as no collisions or instability are triggered in simulation, such plans are incorrectly classified as successful. Figure 6 illustrate several cases where the task goal was clearly unmet, yet no feedback was generated to initiate replanning.

**Fixing the Evaluation Protocol via VLM Check**  To address this fundamental evaluation gap, we introduce a semantic-level verification step using a vision-language model (GPT-4o). After executing each plan, we render the final scene and prompt the VLM to assess whether the natural language goal has been achieved. If not, the failure is recorded and propagated, triggering a replanning cycle. **Specifically, we explicitly instruct the VLM to evaluate structured visual conditions and provide clear success criteria along with positive and negative examples, enabling it to produce consistent and grounded success judgments.**

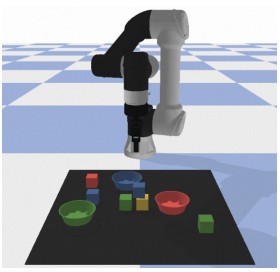 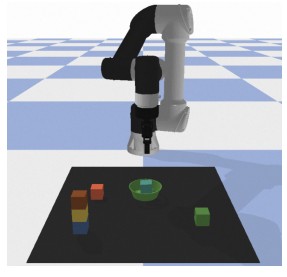 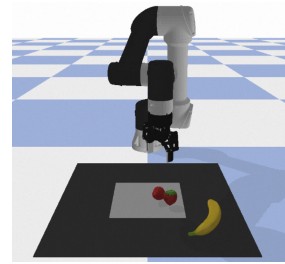

(a) **Pyramid**: the top block does not make contact with the lower-left block to form a pyramid.

(b) **Unstack**: an incorrectly colored block was placed into the target bowl.

(c) **Packing**: there is a object left outside the designated container area.

Figure 6: Task failures caused by incorrect plans that are accepted as successful by PRoC3S, despite violating the task goal, because they do not trigger any constraint violations.

This fix does not modify the core PRoC3S planning mechanism, but augments the evaluation logic with a reliable, goal-aware success signal. It enables all baselines—including PRoC3S and InstructFlow—to be assessed under a consistent, semantically meaningful criterion. While this change may lower the reported success rates of prior methods, we consider it essential for fair and rigorous comparison, especially in tasks with under-specified goals or ambiguous execution semantics.

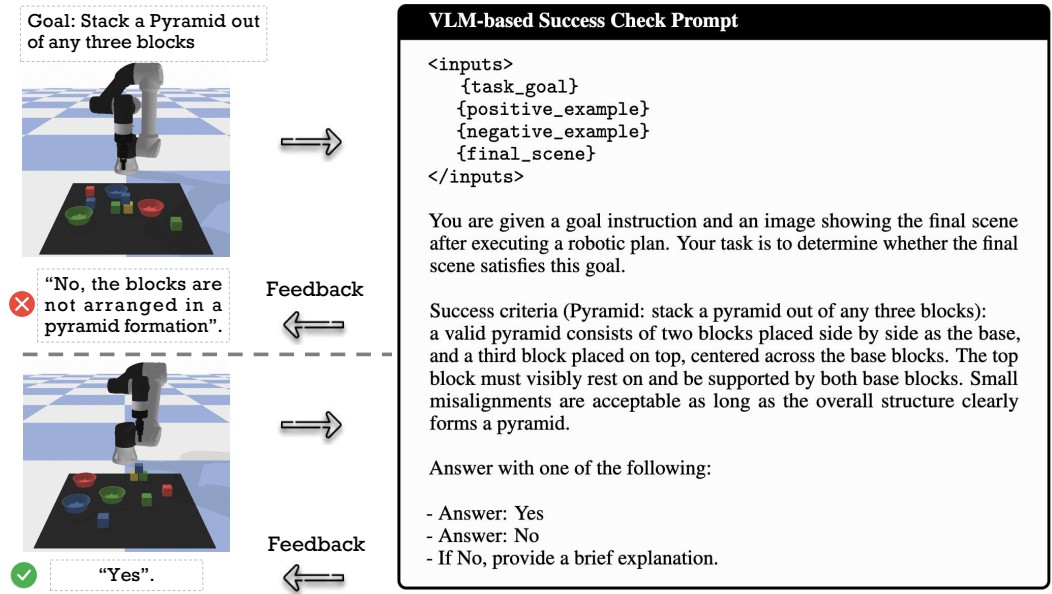

Figure 7: VLM-based success check on two execution plans for the Pyramid task. The upper plan produces a **non-standard pyramid** where the top block does not contact the left base block. The VLM detects this semantic error and returns **"No"** with reason. The lower plan yields a **correct pyramid** structure, and the VLM responds with **"Yes"**. This demonstrates the importance of incorporating VLM check as a complementary layer to evaluation protocol.

## B.2 Additional Experiment Results and Cases

### B.2.1 Additional Experiment

To assess the potential of vision-language models (VLMs) for improving task planning, we conduct an additional experiment where our existing GPT-4o-based reasoning agent receive both the goal and an image of the initial scene, enabling multimodal visual reasoning.

This vision-augmented setup improves the planner's ability to reason under partial observability or ambiguous symbolic states. As shown in Table 5, we observe performance gains in several visually grounded tasks, including *Packing (Blocks)*, *Unstack*, and *Packing (YCB)*, where early-stage visual cues help the system make better object selection or ordering decisions.

| | Drawing | | | | Arrange Blocks | | | | Arrange YCB | |
|---|---|---|---|---|---|---|---|---|---|---|
| | Star | Arrow | Letters | Enclosed | Pyramid | Line | Packing | Unstack | Packing | Stacking |
| LLM[3] (w/ vision) | 40% | 40% | 80% | 50% | 0% | 40% | 30% | 0% | 0% | 10% |
| CaP (w/ vision) | 10% | 0% | 40% | 30% | 20% | 20% | 20% | 20% | 40% | 10% |
| PRoC3S (w/ vision) | 90% | 80% | 80% | 90% | 60% | 70% | 60% | 80% | 40% | 40% |
| InstructFlow (w/ vision) | **100%** | **80%** | **100%** | **100%** | **90%** | **100%** | **90%** | **100%** | **60%** | **70%** |

Table 5: Task success rates (%) across drawing, block arrangement, and YCB manipulation tasks using VLM-based visual reasoning. Bold indicates the best-performing results.

To further evaluate the planning efficiency of InstructFlow, we compare it with the baseline system PRoC3S in terms of (1) average number of feedback queries required per task and (2) end-to-end wall-clock latency.

**Feedback Efficiency.** As shown in Table 6, InstructFlow significantly reduces the number of feedback-driven repair cycles. Tasks that require 2–5 full regenerations in PRoC3S are typically completed in 1-3 symbolic repair steps in InstructFlow, resulting in a reduction of feedback interactions by approximately 37% on average.

| | Drawing | | | | Arrange Blocks | | | | Arrange YCB | |
|---|---|---|---|---|---|---|---|---|---|---|
| | Star | Arrow | Letters | Enclosed | Pyramid | Line | Packing | Unstack | Packing | Stacking |
| PRoC3S | 0.50±0.20 | 1.00±0.20 | 0.60±0.30 | 0.50±1.50 | 1.90±2.25 | 1.20±1.32 | 1.80±2.10 | 2.80±2.22 | 3.10±3.04 | 2.90±2.70 |
| InstructFlow | 0.00±0.00 | 1.00±0.20 | 0.10±0.20 | 0.00±0.00 | 1.30±0.82 | 0.20±0.30 | 1.50±1.22 | 1.90±1.82 | 2.30±2.12 | 2.00±2.10 |

Table 6: Average number of feedback queries per task (mean ± std).

**Wall-clock Latency.** Despite employing a multi-agent planning architecture, InstructFlow does not incur higher computational cost. As shown in Table 7, it achieves an average end-to-end latency reduction of approximately 4.7%. This improvement is attributed to symbolic constraints that prune infeasible code paths and reduce retry cycles in LLM sampling.

| | Drawing | | | | Arrange Blocks | | | | Arrange YCB | |
|---|---|---|---|---|---|---|---|---|---|---|
| | Star | Arrow | Letters | Enclosed | Pyramid | Line | Packing | Unstack | Packing | Stacking |
| PRoC3S | 22.39±3.65 | 24.18±3.77 | 31.16±5.90 | 39.23±53.62 | 152.79±124.54 | 52.64±40.97 | 197.14±210.38 | 142.82±142.34 | 988.23±940.72 | 61.36±19.78 |
| InstructFlow | 20.22±4.12 | 24.20±3.58 | 26.27±4.10 | 36.21±49.25 | 140.93±131.37 | 48.21±37.53 | 204.76±227.24 | 124.30±121.44 | 892.18±880.84 | 63.27±25.51 |

Table 7: Wall-clock latency (in seconds, mean ± std) per task.

These results show that the modular design of InstructFlow not only improves feedback efficiency but also retains low computational latency, making it scalable for complex tasks involving multiple agents and symbolic constraints.

### B.2.2 Table of Symbolic Constraints Discovered

Table 8 summarizes the symbolic constraints that were automatically induced based on failure feedback across different manipulation tasks. From a content perspective, we highlight the following key properties:

**Structural Consistency.** All induced constraints conform to the symbolic constraint formulation presented in Section 3.3. Each constraint instance can be expressed either as a relational predicate $R_i(e_a, e_b)$ or a physical threshold condition $f_j(\Theta) \oplus \tau$. This ensures that all constraints are logically composable, interpretable, and grounded in the formal symbolic space defined by $\phi$.

**Diverse Coverage of Constraint Types.** The constraint set spans a wide range of task-relevant constraint categories, including:

- **Spatial safety:** `ProximitySafe`, `ClearOf`, `PathClear`...
- **Placement feasibility:** `PlacementFeasible`, `Aligned`, `StableOn`...
- **Geometric parameters:** `Distance`, `Offset`, `CenterOfMass`...
- **Temporal logic:** `Before`, `Order`

These categories capture both physical feasibility and symbolic reasoning failure modes, supporting a broad range of corrective strategies.

Notably, the table does not include constraints for the Drawing tasks. This is because the drawing tasks are comparatively simpler in structure and were typically solved by the planner in a single attempt without triggering any failure-driven refinement. As a result, no symbolic constraint induction process was invoked for these tasks, and they are therefore excluded from this table.

| Task | Symbolic Constraints |
|------|----------------------|
| Pyramid | StableOn(?block_top, ?block_bottom); 
 ProximitySafe(?block_bottom); 
 PathClear(?gripper, ?block); 
 Before(place(block_top), place(?block_bottom)); 
 Distance(?block_bottom) $\in [0, 0.04]$ 
 Aligned(?block_bottom) $\wedge$ On(block_top, ?block_bottom); 
 Order(Place(block_bottom1), Place(block_bottom2), Place(block_top)); 
 Contact(block_top, ?block_bottom) $\wedge$ CloseTogether(?block_bottom); 
 CenterOfMass(block_top) $\in$ SupportArea(?block_bottom) |
| Line | StableOn(?block, ?table); 
 Alignment(?block); 
 Alignment_tolerance $\in [-0.01, 0.01]$ |
| Packing (Blocks) | ProximitySafe(?block, ?boundary); 
 PlacementFeasible(?block, square_region); 
 WithinBounds(?block, region_center, 2*blcok_size); 
 ProximitySafe(block, region_center) $\wedge$ 
 WithinDistance(?block, region_center, ?max_distance) |
| Unstack (Blocks) | StableOn(?block, ?bowl); 
 ProximitySafe(?gripper, ?block); 
 Clearof(?gripper,?obstacle); 
 PlacementFeasible(green_block, green_bowl); 
 Inside(green_block, green_bowl); 
 Aligned(block_center, bowl_center); 
 NotStacked(green_block, ?obstacle) $\wedge$ offset_z $> 0.03$ |
| Packing (YCB) | ProximitySafe(?gripper, ?object); 
 GraspFeasible(?grasp, ?object_pose); 
 ProximitySafe(object, table_center); 
 Distance(?object, table_center) $< 0.06$; 
 PlacementFeasible(?object, center, threshold) $\wedge$ threshold $= 0.06$; 
 Graspable(?grasp, ?object); 
 CollisionFree(?gripper, ?object) |
| Stacking (YCB) | OnTop(object_a, object_b); 
 ClearOf(object_a, surface); 
 ClearSurface(object_b); 
 PlacementFeasible(object_a, object_b); 
 StableOn(object_a, object_b); 
 AlignedForStacking(object_a, object_b); 
 Graspable(?object, ?grasp) $\wedge$ CollisionFree(?object) |

Table 8: Inducted constraints for each task across Arrange-Blocks and Arrange-YCB domains

### B.2.3 Case study

In addition to the `Unstack` task discussed in section 4.4, we present several representative manipulation tasks as case studies to further demonstrate the effectiveness and generality of our method. For each task, we analyze the core planning difficulties, the symbolic constraints induced during execution, and how InstructFlow dynamically updates the instruction graph to recover from failures.

**Pyramid.** The goal is to **stack a Pyramid out of any three blocks**, which means the robot need to construct a pyramid-like structure by selecting any three available blocks: two as the base and one stacked on top. The key challenge lies in both object selection and precise spatial configuration. Specifically, the lateral distance between the two base blocks must be carefully chosen to ensure that the top block can be stably placed across them. Furthermore, this task is especially sensitive to execution noise: even if a plan passes all physical constraint checks in simulation, the same stack may collapse in the real environment due to minor perturbations such as control inaccuracy or object pose

estimation errors. PRoC3S [3] lacks a feedback mechanism to detect post-simulation failures. Once a plan passes simulation, it is executed directly without verifying whether the real-world outcome satisfies the goal. Our method addresses this by introducing a VLM-based validation step: the executed scene is rendered and checked against the original goal, with replanning initiated if the structure is incorrect.

In the first round of code generation, InstructFlow receives the goal: *"Stack a pyramid out of any three blocks."* The **InstructFlow Planner** constructs an initial instruction graph consisting of seven planning nodes, each representing a subgoal in the pyramid assembly process. Guided by this graph, the code generator produces an executable plan. However, after executing the plan, VLM-based semantic validation reports a failure, indicating the following issue:

---
**[Error Message]**: "The blocks are not arranged in a pyramid formation"

---

Given the failure feedback and generated code, the **Constraint Generator** localizes the issue to the code block responsible for pyramid construction, specifically the subgoal corresponding to `subgoal5` in `get_plan()` and the `offset_x` parameter defined in `get_domain()`. By examining the sampled parameter values, InstructFlow infers that the current range of `offset_x` is too wide, causing the base blocks to be placed too far apart. As a result, the top block either falls during execution or forms a configuration that is not recognized as a pyramid by the VLM. This reasoning leads to the generation of an explicit symbolic constraint:

---
$\phi_{\text{dist}} := \texttt{Distance}(\texttt{?block\_bottom}) \in [0, 0.04]$

---

Here, 0.04 corresponds to the environment-defined block size, ensuring that the top block can span both base blocks without falling or misalignment.

Given the symbolic constraint, **InstructFlow Planner** dynamically updates the instruction graph by introducing reasoning nodes that refine the value range of `offset_x`. Specifically, the updated graph incorporates a `parameter range refinement node` $v_{\mathcal{T}param}^{\text{reason}}$, whose output is an instruction to narrow the sampling bounds of `offset_x`.

Based on the updated instruction graph, the **Code Generator** regenerates the executable code to satisfy the induced symbolic constraints. As shown in Figure 5, the regenerated code modifies the `get_domain()` function by narrowing the sampling range of `offset_x` from the original (`BLOCK_SIZE`, `BLOCK_SIZE * 2`) to (`0`, `BLOCK_SIZE`). This adjustment results in a tighter arrangement of the base blocks, enabling the top block to rest stably and form a recognizable pyramid structure. The final scene passes VLM validation, and the task is successfully completed.

**Stacking (Arrange-YCB).** This task requires the agent to stack one object on top of another, selected from a diverse set of YCB objects with varied shapes, sizes, and physical properties. The primary challenge lies in object selection: due to irregular geometries and asymmetric mass distributions, not all object pairs are feasible for stable stacking. Therefore, selecting an appropriate pair is critical to the success of the task. A second challenge arises from the lack of feedback handling in PRoC3S. Due to shape mismatch or small execution disturbances, the stacked configuration may fail in the real environment but without replanning.

In the first round of code generation, InstructFlow receives the goal: "Stack any object on any other object". The **InstructFlow Planner** constructs an initial instruction graph consisting of three planning nodes, corresponding to selecting two objects, picking the first object, and placing it on top of the second. Guided by this structure, the **Code Generator** generates an executable plan. However, after execution, VLM-based semantic validation reports a failure, indicating that the chosen object pair did not result in a valid or stable stacked configuration.

---
**[Error Message]**: "No objects are stacked on top of each other"

---

Given the failure feedback and generated code, the **Constraint Generator** localizes the issue to the code block responsible for selecting object pairs. Specifically, the analysis reveals that the object

```
1    def gen_plan(initial: RavenBelief,
2                 base_x: float, base_y: float,
3                 offset_x: float):
4        plan = []
5
6        # Subgoal 1: Select three blocks for the pyramid
7        blocks = [obj for obj in initial.objects.values() if obj.category == "block"]
8        block1, block2, block3 = blocks[:3]
9
10       # Subgoal 2: Pick the first block for the base of the pyramid
11       plan.append(Action("pick", block1.pose.point))
12
13       # Subgoal 3: Place the first block at the base position
14       plan.append(Action("place", [base_x, base_y, block1.pose.point[2]]))
15
16       # Subgoal 4: Pick the second block for the base of the pyramid
17       plan.append(Action("pick", block2.pose.point))
18
19       # Subgoal 5: Place the second block next to the first block at the base position, ensuring alignment
20       plan.append(Action("place", [base_x + offset_x, base_y, block2.pose.point[2]]))
21
22       # Subgoal 6: Pick the third block for the top of the pyramid
23       plan.append(Action("pick", block3.pose.point))
24
25       # Subgoal 7: Place the third block on top of the first and second blocks to complete the pyramid
26       plan.append(Action("place", [base_x + offset_x / 2, base_y, block1.pose.point[2] + BLOCK_SIZE]))
27
28       return plan
29
30   def gen_domain(initial: RavenBelief):
31       return {
32           "base_x": ContinuousSampler(TABLE_BOUNDS[0][0], TABLE_BOUNDS[0][1] - BLOCK_SIZE),
33           "base_y": ContinuousSampler(TABLE_BOUNDS[1][0], TABLE_BOUNDS[1][1]),
34  -        "offset_x": ContinuousSampler(BLOCK_SIZE, BLOCK_SIZE * 2),
34  +        "offset_x": ContinuousSampler(0, BLOCK_SIZE),
35       }
```

Figure 8: A code snippet illustrating how InstructFlow repairs the **Pyramid** task plan: to correct the code, **InstructFlow** incorporates a **parameter range refinement node** into the instruction graph based on the induced symbolic constraint, which guides the regenerated code to reduce the sampling range of **offset_x** in the **get_domain** function. This adjustment reduces the spacing between base blocks, enabling a **stable top placement** and successful pyramid construction.

pair was selected via an environment-defined `DiscreteSampler()`, which randomly samples two objects without considering any factors that influence stackability. As a result, the constraint generator induces the following symbolic constraint to guide future selection:

$$\phi_{\text{stack}} := \texttt{AlignedForStacking(object\_a, object\_b)} \wedge \\ \texttt{PlacementFeasible(object\_a, object\_b)}$$

Given the symbolic constraint, **InstructFlow Planner** dynamically updates the instruction graph by introducing reasoning nodes that guide the selection of a physically compatible object pair for stacking. Specifically, the updated graph includes a `spatial_relation_reasoning_node` $v_{\mathcal{T}spatial}^{\text{reason}}$ to analyze relative object positions and an `object_density_analysis_node` $v_{\mathcal{T}density}^{\text{reason}}$ to evaluate the local clutter surrounding each object. These outputs are fed into an `object_selection_reasoning_node` $v_{\mathcal{T}select}^{\text{reason}}$, which selects `object_2` and `object_4` as a feasible stacking pair based on the symbolic constraint `SelectStackablePair`. A `plan_logic_reasoning_node` $v_{\mathcal{T}plan}^{\text{reason}}$ then determines the appropriate execution order, generating a subgoal sequence: "Pick `object_2`" followed by "Place `object_2` on top of `object_4`.".

Based on the updated instruction graph, the **Code Generator** regenerates the executable code to satisfy the induced symbolic constraints. As shown in Figure 6, the regenerated code eliminates the use of unconstrained object sampling in `gen_domain()`, replacing it with fixed object assignments (`object_2` and `object_4`) determined through reasoning nodes. This ensures that the selected object pair adheres to the `AlignedForStacking` and `PlacementFeasible` constraints. The new plan simplifies the domain by sampling only the grasp for the manipulated object and directly encodes the stacking intent in the plan logic. With this targeted refinement, the resulting scene passes VLM-based check, and the stacking task is successfully completed.

```
 1   -     def gen_plan(initial: RavenBelief, object_name1: str, grasp1: RavenGrasp,
 2                       object_name2: str, grasp2: RavenGrasp):
     1   +     def gen_plan(initial: RavenBelief, grasp: RavenGrasp):
 3
 4             plan = []
 5
 6             #Subogoa 1: Pick the first selected object
 7             plan.append(Action("pick", [object_name1, grasp1]))
 8
 9             #Subogoa 2: Get the pose of the second object to stack on top
10   -         second_object_pose = initial.objects[object_name2].pose
11
12             #Subogoa 3: Place the first object on top of the second object
13             plan.append(Action("place", [object_name1, grasp1, RavenPose(x=second_object_pose.x,
14                 y=second_object_pose.y, z=second_object_pose.z + BLOCK_SIZE)]))
15
     2             #Subogoa 1: Define the objects to be used in the plan
     3             object_name1 = "object_2"  # apple
     4             object_name2 = "object_4"  # power_drill
     5
     6   +         #Subogoa 2: Pick the first selected object (apple)
     7             plan.append(Action("pick", [object_name1, grasp]))
     8
     9             #Subogoa 3: Get the pose of the second object (power_drill) to stack on top
    10             second_object_pose = initial.objects[object_name2].pose
    11
    12             #Subogoa 4: Place the first object (apple) on top of the second object (power_drill)
    13             plan.append(Action("place", [object_name1, grasp, RavenPose(x=second_object_pose.x,
    14                 y=second_object_pose.y, z=second_object_pose.z + BLOCK_SIZE)]))
16           return plan
17
18       def gen_domain(initial: RavenBelief):
19           object_ids = list(initial.objects.keys())
20           return {
21   -           "object_name1": DiscreteSampler(object_ids),
22             "grasp1": GraspSampler(),
23   -           "object_name2": DiscreteSampler(object_ids),
24             "grasp2": GraspSampler(),
25           }}
    15   +       return {
    16               "grasp": GraspSampler(),
    17             }
```

Figure 9: A code snippet illustrating how InstructFlow repairs the **Stacking** task plan: to correct the code, **InstructFlow** updates the instruction graph based on the induced symbolic constraint. This update adjusts the subgoals from **randomly selecting stacking targets** to using a **specifically determined object pair**, guiding the regenerated code to modify the corresponding logic in **get_plan** and remove the random sampling from **get_domain**. This adjustment ensures the **physical feasibility** of the stacking operation.

