# OpenReview forum: "InstructFlow: Adaptive Symbolic Constraint-Guided Code Generation for Long-Horizon Planning"
_NeurIPS.cc/2025/Conference — NeurIPS 2025 poster_

### Official Review · Reviewer_iwsg · 2025-07-01

**Clarity:** 2
**Significance:** 2
**Originality:** 3
**Rating:** 5
**Confidence:** 3

**Summary:**

InstructFlow is a symbolic, feedback-driven framework for robotic code generation that decomposes tasks using a hierarchical instruction graph. When execution failures occur, a constraint generator analyzes the feedback to infer symbolic constraints that identify the root cause of the problem. These constraints then guide targeted, efficient code repairs without regenerating the entire plan from scratch. This method significantly enhances robustness and task success rates in complex, long-horizon manipulation scenarios.

**Questions:**

1. Regarding the limitations of InstructFlow, what analysis has been conducted on its failure cases or the scenarios in which its performance is suboptimal? Furthermore, which component within the framework (say, the Symbolic Constraint Generator) is most frequently the point of failure?
2. Have metrics such as wall-clock computation time and the average number of feedback queries been benchmarked for InstructFlow? If so, what are the quantitative improvements in planning efficiency and query complexity relative to the PROC3S baseline?

**Ethical Concerns:**

["NO or VERY MINOR ethics concerns only"]

**Final Justification:**

I believe the approach provides several useful contributions that enhance prior LLM-based robotic planning frameworks (viz. PRoC3S) with better symbollic constraints and informed re-planning.

**Limitations:**

- The framework's efficacy is reduced in scenarios requiring precise manipulation of objects with irregular geometries as evidenced by the suboptimal success rates in the Arrange-YCB domain, which were 60% for packing and 70% for stacking, respectively.
- Dependency on closed LLM models accessed via a cloud API. There is a lack of analysis of the framework's viability with open-source or locally hosted models, which is critical for applications in autonomous systems where network-independent, real-time processing is often a requirement.

**Quality:**

3

**Strengths And Weaknesses:**

## Strengths:

- Extends the PRoC3S [3] robot planning framework with symbolic reasoning features and constraint repair functionality.
- More efficient and robust than PROC3S's reactive  method of starting over, which lacks mechanisms to reason about the task-level causes of failures. The system uses a graph as an internal plan representation that can be adjusted mid-stream.
- This focus on generalization is a step towards more intelligent and less brittle robotic systems. Instead of solving just the immediate problem, the system learns a more fundamental rule which can help LLM planners to correctly refine the plan.

## Weaknesses:

- The symbolic inference phase of the constraint generator relies on unverified outputs. If these are incorrectly identified, the plan repair would most likely be incorrect.
- Lack of real-world experiments evaluating the sim-to-real gap.

### Typos:

- L105 a LLMis
- L627: offse_x

---

> ### Author Rebuttal · Authors · 2025-07-31
>
> Thank you for your thoughtful review and encouraging assessment. We deeply appreciate your support and careful reading of our work. Below, we respond to your comments with detailed clarifications and highlight the revisions incorporated based on your suggestions.
>
> ### Weakness 1:
>
> > “The symbolic inference phase of the **constraint generator relies on unverified outputs**. If these are incorrectly identified, the plan repair would most likely be incorrect.”
>
>
> **Response**:
>
> We appreciate the reviewer’s concern. While symbolic inference can in principle introduce errors, InstructFlow mitigates this via both structured reasoning and semantic validation:
> - **Grounded diagnostic pipeline:** As detailed in **`Section 3.2`** , symbolic constraints are extracted through a four-step diagnostic workflow: from *failure entity retrieval* to *code-level and diagnostic analysis*, rather than shallow pattern matching.  This structured pipeline mitigate hallucination risk.
> - **Empirical Support:** `Ablation result` from **`Table 2(Lines 283-284)`** shows a 30-40% drop without Constraint Generator; case studies (`Appendix B.2.3`) and a illustration of constraints discovered in **`Table 4`**`(Lines 611-612)` confirm the alignment of induced constraints with task semantics.
>
> ---
>
> ### Weakness 2:
>
> > “*Lack of **real-world experiments evaluating the sim-to-real** gap.*”
> >
>
> **Response**:
>
> We agree that real-robot validation is valuable. While our study is simulation-based, we explicitly model two key real-world uncertainties: **sensor noise** and **feedback ambiguity**, to evaluate robustness:
>
> 1. **Perceptual noise:** We inject Gaussian perturbations into object poses, $\sigma\in\{0.005,0.01,0.02\}$. Note that object size is 0.04, the max noise is up to 50% object size. As shown in Table 1, InstructFlow maintains >70% success across tasks even under perceptual noise.
>
>
> |   Table 1 (Reviewer-iwsg)     |  Pyramid | Line | Packing | Unstack | YCB-Packing | YCB-Stacking | Averge Drop    |
> | --------------------- | ------- |:----:| ------- | ------- | ------- | -------- | --- |
> | InstructFlow          | 90%     | 100% | 90%     | 90%     | 60%     | 70%      |   -  |
> | InstructFlow $\sigma=$ 0.005  | 90%     | 100% | 90%     | 90%     | 60&     | 70%      |   0%  |
> | InstructFlow $\sigma=$ 0.01   | 80%     | 100% | 80%     | 80%     | 50%     | 60%      |  8.33%   |
> | InstructFlow $\sigma=$ 0.02   | 70%     | 90%  | 70%     | 70%     | 40%     | 50%      |  18.33%   |
>
> 2. **Feedback noise:** We simulate **incorrect** and **incomplete** failure feedbacks.  (1) incorrect feedbacks with wrong object references, and (2) incomplete traces with missing object IDs or causes. Table 2 shows InstructFlow remains robust, with only 11–16% average drop despite noisy or missing feedback.
>
>
> | Table 2 (Reviewer-iwsg)  |Star | Arrow | Letters | Enclosed | Pyramid | Line | Packing | Unstack | YCB-Packing | YCB-Stacking | Average Drop    |
> | -------------------------------- | ---- |:-----:| ------- | -------- | ------- | ---- | ------- | ------- | ------- | -------- | --- |
> | InstructFlow                     | 100% |  80%  | 100%    | 100%     | 90%     | 100% | 90%     | 90%     | 60%     | 70%      |  -   |
> | InstructFlow w/incorrect feedback  | 100% |  80%  | 100%    | 100%     | 60%     | 80%  | 70%     | 60%     | 30%     | 40%      |   16%  |
> | InstructFlow w/incomplete feedback | 100% |  80%  | 100%    | 100%     | 70%     | 90%  | 70%     | 70%     | 40%     | 50%      |   11%  |
>
> These results show InstructFlow operates reliably without perfect state or supervision，a key requirement for real-world deployment. We view hardware validation as important future work and believe these robustness results provide strong evidence of real-world readiness.
>
> ---
>
> ### Question 1:
>
> > “Regarding the limitations of InstructFlow, **what analysis has been conducted on its failure cases or the scenarios in which its performance is suboptimal**? Furthermore, which component within the framework (say, the Symbolic Constraint Generator) is most frequently the point of failure?”
> >
>
> **Response**:
>
> We thank the reviewer for raising this important point! Our analysis indicates that InstructFlow performs suboptimally in tasks with *high object diversity*, *irregular geometries*, or *tight layout constraints*, notably `Packing-YCB` and `Stacking-YCB` (`Fig 5, Lines 636-642 from Appendix B.2.3`). These scenarios challenge both constraint induction and plan refinement, due to complex spatial couplings and ambiguous success conditions.
>
> For instance:
> 1. **In Packing-YCB,** example shows how **constraints that require spatial precision** (like placement feasibility) are hard to induce when object shapes are varied. This clarifies **why performance degrades** in this task, a failure mode tied to **constraint induction under geometry noise**.
> 2. **In Stacking-YCB,** example emphasizes failures from reasoning about **geometric compatibility and object stability**, again showing a case where **symbolic reasoning and plan refinement** can fail due to ambiguity in object interactions.
>
> At the component level, the **Code Generator** emerges as the most common failure point. To support this, we conducted ablation experiments (Table 3, Reviewer-iwsg) by replacing GPT-4o with GPT-3.5-turbo across modules. The steepest performance drop (13%) occurs in the Code Generator, compared to 8% in the Constraint Generator and 4% in the Planner, highlighting the Code Generator’s sensitivity in long-horizon, constraint-heavy tasks.
>
> |  Table 3  (Reviewer-iwsg)|Star | Arrow | Letters | Enclosed | Pyramid | Line | Packing | Unstack | YCB-Packing | YCB-Stacking |  Average Drop   |
> | ---------------------------------------- | ---- |:-----:| ------- | -------- | ------- | ---- | ------- | ------- | ------- | -------- | --- |
> | InstructFlow w/**GPT-4o**                | 100% |  80%  | 100%    | 100%     | 90%     | 100% | 90%     | 90%     | 60%     | 70%      | -  |
> | InstructFlow w/**GPT-3.5-turbo**         | 90%  |  70%  | 90%     | 100%     | 60%     | 80%  | 70%     | 60%     | 40%     | 50%      |   17% $\downarrow$   |
> | Code generator w/**GPT-3.5-turbo**       | 90%  |  70%  | 100%    | 100%     | 70%     | 80%  | 70%     | 60%     | 50%     | 60%      |    13% $\downarrow$   |
> | Constraint generator w/**GPT-3.5-turbo** | 90% |  80%  | 100%    | 100%     | 70%     | 90% | 90%     | 70%     | 50%     | 60%      |   8% $\downarrow$  |     |
> | Planner w/**GPT-3.5-turbo**              | 90% |  80%  | 100%    | 100%     | 80%     | 100% | 90%     | 80%     | 60%     | 60%      |   4% $\downarrow$    |
>
>
> ---
>
> ### Question 2:
>
> > “Have metrics such as **wall-clock computation time** and the **average number of feedback queries** been benchmarked for InstructFlow? If so, what are the **quantitative improvements in planning efficiency** and query complexity relative to the PROC3S baseline?”
> >
>
> **Response**:
>
> We appreciate the reviewer’s interest in benchmarking efficiency and query complexity.
> While InstructFlow introduces modular agents, it achieves superior **planning efficiency** and **comparable or improved latency** relative to PRoC3S, as shown below.
>
>
> 1. **Fewer feedback queries:** As summarized in **Table 4**, InstructFlow reduces feedback-driven repair cycles by **~37%** on average. For example, in chellenging tasks like `Unstack` and `Pyramid`, InstructFlow completes planning in just 1–3 repair steps, compared to 2–5 full regenerations in PRoC3S. This gain stems from localized plain repair and high-level symbolic guidance.
>
> | Table 4 (Reviewer-iwsg)  | Star |Arrow | Letters | Enclosed | Pyramid | Line  |  Packing   |  Unstack   |   Packing  |  Stacking   |
> | ------------------ | ----------- | ---------- |:-----------:| ----------- | ----------- | ----------- | ----------- | ----------- | ----------- | ----------- |
> | PRoC3S             | 0.50±0.20  |1.00±0.20 | 0.60±0.30 | 0.50±1.50 | 1.90±2.25 | 1.20±1.32 | 1.80±2.10  | 2.80±2.23 | 3.10±3.04 | 2.90+2.70 |
> | InstructFlow(ours) | 0.00±0.00      |     1.00±0.20        |    0.10±0.20      |      0.00±0.00        |      1.30+0.82       |      0.20+0.30       |       1.50+1.22      |       1.90+1.82      |     2.30+2.12        |      2.00+2.10       |
>
> 2. **Lower wall-clock latency:** Despite its multi-agent design, InstructFlow achieves a **~4.7% reduction** in end-to-end latency (see **Table 5** below). Symbolic constraints (e.g., `Distance(?block_bottom) ∈ [0, 0.04]`) help prune infeasible code regions, reducing LLM sampling and retry overhead.
>
> | Table 5 (Reviewer-iwsg)  | Star |Arrow | Letters | Enclosed | Pyramid | Line  |  Packing   |  Unstack   |  YCB-Packing  |  YCB-Stacking   |
> | ------------------ | ----------- | ---------- |:-----------:| ----------- | ----------- | ----------- | ----------- | ----------- | ----------- | ----------- |
> | PRoC3S             | 22.39±3.65  | 24.18±3.77 | 31.16±5.90  | 39.23±53.62 | 152.79±124.54 | 52.64±40.97 | 197.14±210.38  | 142.82±142.34 | 988.23±940.72 | 61.36±19.78 |
> | InstructFlow(ours) |    20.22+4.12     |       24.20+3.58     |   26.27+4.10      | 36.21+49.25            |      140.93+131.37       |      48.21+37.53       |       204.76+227.24      |     124.30+121.44        |      892.18+880.84       |      63.27+25.51       |
>
> These improvements were obtained without latency-specific tuning, suggesting that modular planning does not incur significant computational penalties. We will report full statistics in the final version and are happy to release detailed logs upon request.
>
> ---
>
> ### Writing issues:
>
> Thank you for spotting these typos. We will thoroughly proofread and fix all such issues in the final version to ensure clarity and polish.
>
> ---
>
> Once again, We are deeply appreciative of your generous support and constructive feedback. Your review not only affirmed our direction but also inspired valuable refinements.

---

> > ### Comment · Reviewer_iwsg · 2025-08-06
> >
> > I thank the authors for their detailed rebuttal. My primary concern regarding the potential for unverified outputs from the symbolic constraint generator has been effectively addressed. The explanation of the structured, four-stage diagnostic pipeline clarifies that the symbolic induction is not arbitrary. This is further supported by the compelling empirical evidence from the ablation studies, which demonstrate the component's significant positive impact on performance.
> >
> > I also appreciate the additional efficiency comparisons with PROC3S and have no further comments apart from the minor suggestion that the new tables be placed in the appendix. Please also ensure all time-based metrics are clearly labeled with their units (e.g., seconds) for clarity.

---

### Official Review · Reviewer_fD4v · 2025-07-03

**Clarity:** 3
**Significance:** 2
**Originality:** 3
**Rating:** 4
**Confidence:** 4

**Summary:**

The paper addresses the brittleness of large-language-model planners on long-horizon robotic manipulation tasks, where naive code generation often produces physically infeasible plans and, upon execution failures, resorts to blind retries without diagnosing root causes. To overcome this, the authors propose InstructFlow, a multi-agent framework that constructs a hierarchical instruction graph to decompose high-level goals into semantically typed subgoals, embeds a Constraint Generator that analyzes failure traces (e.g., collisions) to induce precise symbolic predicates and inject them back into the graph, and orchestrates a targeted, graph-guided code-generation loop that updates only the affected subgraph rather than regenerating the entire plan. Empirical results on drawing, block-stacking, and YCB object-manipulation benchmarks demonstrate marked improvements in success rates and robustness compared to existing LLM-based planners.

**Questions:**

Writing issues
- Line 105 in the PDF reads "a LLMis instructed...", missing a space ("LLM is").
- In Figure 1's caption, "Phsical Diagnosis" is a typo, should be "Physical Diagnosis."
- In Figure 1 and Figure 2, "griper" is misspelled; it should read "gripper".

Questions
- How sensitive is InstructFlow's performance to the clarity and structure of the user-written instruction scaffold (e.g., decomposition prompts, action templates)? Have you evaluated how variations or errors in those instructions impact success rates and repair effectiveness?
- Constraint Generator assumes clean, structured failure traces (e.g. precise collision logs), but real‐world feedback is often noisy or incomplete, how does InstructFlow handle ambiguous or incorrect feedback data?

**Ethical Concerns:**

["NO or VERY MINOR ethics concerns only"]

**Final Justification:**

The revision demonstrates meaningful progress in addressing key concerns, particularly around logical coherence and experimental support. However, the scope of comparison remains somewhat narrow.

**Limitations:**

yes.

**Paper Formatting Concerns:**

No paper formatting concerns.

**Quality:**

2

**Strengths And Weaknesses:**

Pros
- InstructFlow achieves substantial gains in task success(20-40 percentage points over strong baselines)across drawing, block-stacking, and YCB benchmarks, demonstrating its robustness in long-horizon planning under physical constraints.
- The hierarchical instruction graph cleanly decomposes high-level goals into typed subgoals, providing clear semantic and spatial structure for both initial plan synthesis and targeted repair.
- Its automatic symbolic constraint induction converts raw failure traces (e.g. collisions) into reusable predicates that guide focused code refinement, no per-task hand-crafting of constraints is needed.

Cons
- The multi-agent, graph-guided pipeline incurs extra LLM calls and bookkeeping (planning, code generation, constraint induction), leading to higher computational overhead and latency compared to flat planning methods.
- Setting up the domain requires defining action templates, object/region vocabularies, and API mappings-a nontrivial engineering effort that may slow adoption in new environments.
- In scenarios with simple, short-horizon tasks, the marginal benefit of structured repair can be limited, suggesting diminishing returns when failures are rare.

---

> ### Author Rebuttal · Authors · 2025-07-31
>
> We would like to express our sincere respect for your insightful review! In the following, we systematically respond to your concerns with additional analysis and empirical results where appropriate.
>
>
> ### Weakness 1:
> >*The multi-agent, graph-guided pipeline incurs **extra LLM calls and bookkeeping** (planning, code generation, constraint induction), leading to **higher computational overhead** and **latency** compared to flat planning methods.*
>
>
> **Response:**
>
> While our pipeline introduces modular components, InstructFlow achieves **lower latency** and **fewer LLM calls** compared to flat planners like PRoC3S[1]:
>
> 1. **Efficiency**: InstructFlow reduces feedback rounds by **37%** on average (Table 1), often converging in **1–3 ** vs. **2–5 ** regenerations in PRoC3S across complex tasks.
>
> |Table1 (Reviewer-fD4v )  | Star |Arrow | Letters | Enclosed | Pyramid | Line  |  Packing   |  Unstack   |   YCB-Packing  |  YCB-Stacking   |
> | ------------------ | ----------- | ---------- |:-----------:| ----------- | ----------- | ----------- | ----------- | ----------- | ----------- | ----------- |
> | PRoC3S             | 0.50±0.20  |1.00±0.20 | 0.60±0.30 | 0.50±1.50 | 1.90±2.25 | 1.20±1.32 | 1.80±2.10  | 2.80±2.23 | 3.10±3.04 | 2.90+2.70 |
> | InstructFlow(ours) | 0.00±0.00      |     1.00±0.20        |    0.10±0.20      |      0.00±0.00        |      1.30+0.82       |      0.20+0.30       |       1.50+1.22      |       1.90+1.82      |     2.30+2.12        |      2.00+2.10       |
>
> 2. **Wall-clock time**: As shown in Table 2 below, InstructFlow reduces latency by **~4.7%**, benefiting from symbolic constraints that narrow the search space and reduce code sampling needs.
>
> | Table2 (Reviewer-fD4v)   | Star |Arrow | Letters | Enclosed | Pyramid | Line  |  Packing   |  Unstack   |   YCB-Packing  |  YCB-Stacking   |
> | ------------------ | ----------- | ---------- |:-----------:| ----------- | ----------- | ----------- | ----------- | ----------- | ----------- | ----------- |
> | PRoC3S             | 22.39±3.65  | 24.18±3.77 | 31.16±5.90  | 39.23±53.62 | 152.79±124.54 | 52.64±40.97 | 197.14±210.38  | 142.82±142.34 | 988.23±940.72 | 61.36±19.78 |
> | InstructFlow(ours) |    20.22+4.12     |       24.20+3.58     |   26.27+4.10      | 36.21+49.25            |      140.93+131.37       |      48.21+37.53       |       204.76+227.24      |     124.30+121.44        |      892.18+880.84       |      63.27+25.51       |
>
> [1] Trust the proc3s: Solving long-horizon robotics problems with llms and constraint satisfaction.(CoRL'2024)
>
> ---
>
>
> ### Weakness 2:
>
> > *Setting up the domain requires **defining action templates, object/region vocabularies, and API mappings**-a nontrivial engineering effort that **may slow adoption in new environments.***
> >
>
> **Response**:
>
> InstructFlow follows the same setup convention as PRoC3S, a one-time domain-level definition reused across tasks. Action templates (e.g., pick/place) are generic and map easily to common robotic APIs, making the system modular and portable to new domains with minimal overhead.
>
> ---
>
> ### Weakness 3:
>
> > *“In scenarios with simple, **short-horizon tasks**, the **marginal benefit of structured repair can be limited**, suggesting diminishing returns when failures are rare.”*
> >
>
> **Response**:
>
> We would like to clarify that **InstructFlow is designed to adapt to task complexity,** and does not incur unnecessary overhead in simple, short-horizon tasks. Specifically:
> 1. **Failure-driven activation:** The structured repair mechanism (constraint generator)  is only triggered when failures occur. In simple or successful cases, it remains dormant, and behaves like a flat planner, incurring no extra overhead and avoiding diminishing returns.s like a flat planner with negligible overhead.
> 2. **Additional benefits:** Even when no failure occurs, InstructFlow’s hierarchical instruction graph provides clear semantic structure to the plan, improving **interpretability** (what subgoals are being pursued), and **debuggability** (where and why a plan might go wrong if it ever does). This structured representation facilitates downstream analysis, monitoring, and integration with safety checks，even in short-horizon tasks.
> In summary, InstructFlow avoids overhead when unnecessary and provides benefits when needed, ensuring that its structural design remains practical across both simple and complex tasks. We will clarify this design behavior more clearly in the revised manuscript.
>
> ---
> ### Question 1:
>
> > *“How **sensitive** is InstructFlow's performance **to the clarity and structure of the user-written instruction scaffold** (e.g., decomposition prompts, action templates)? Have you evaluated how variations or errors in those instructions impact success rates and repair effectiveness?”*
> >
>
> **Response**:
>
> To clarify, InstructFlow does not require users to write any instruction scaffold, decomposition prompt, or action template. **The “instruction scaffold” in our setting refers to a few-shot prompt provided to the Planner Agent,** which contains domain-level decomposition examples. These are fixed across evaluation and selected to be representative of the domain without overlapping with evaluation tasks."
>
> In response to your valuable suggestion, we conduct an ablation study in which **the few-shot prompt is changed from same domain tasks to out-of-domain examples** (e.g., using examples from `Arrange YCB` to guide `Arrange Block`). As shown in Table 3, performance drops moderately but remains competitive, indicating that InstructFlow is reasonably robust to prompt variation.
>
> |  Table3 (Reviewer-fD4v)            | Pyramid | Line | Stacking | Unstack |
> |-----------------|---------|------|----------|---------|
> | InstructFlow             | 90%     | 100% | 90%      | 90%     |
> | InstructFlow w/ YCB-example | 60%     | 80%  | 60%      | 50%     |
>
> ---
>
> ### Question 2:
>
> > *“Constraint Generator assumes **clean, structured failure traces** (e.g. precise collision logs), but real‐world feedback is often noisy or incomplete, how does InstructFlow handle **ambiguous or incorrect feedback data**?”*
> >
>
> **Response**:
>
> To directly address this concern, we conducted two sets of **robustness experiments**:
> 1. **Incorrect Feedback**： the structured trace includes deliberately *incorrect object references* (e.g., reporting a collision with object B when object A was responsible);
> 2. **Incomplete Feedback**： *key information* such as object IDs or failure causes is *removed*, simulating partial or vague real-world signals.
>
> |  Table4 (Reviewer-fD4v)  |Star | Arrow | Letters | Enclosed | Pyramid | Line | Packing | Unstack | YCB-Packing | YCB-Stacking | Average Drop    |
> | -------------------------------- | ---- |:-----:| ------- | -------- | ------- | ---- | ------- | ------- | ------- | -------- | --- |
> | InstructFlow                     | 100% |  80%  | 100%    | 100%     | 90%     | 100% | 90%     | 90%     | 60%     | 70%      |  -   |
> | InstructFlow w/incorrect feedback  | 100% |  80%  | 100%    | 100%     | 60%     | 80%  | 70%     | 60%     | 30%     | 40%      |   16%  |
> | InstructFlow w/incomplete feedback | 100% |  80%  | 100%    | 100%     | 70%     | 90%  | 70%     | 70%     | 40%     | 50%      |   11%  |
>
> Results (Table 4) show minimal degradation, with InstructFlow maintaining near-baseline performance. In addition, InstructFlow already handles unstructured semantic feedback via a VLM-based goal checker (`Lines 544-566, Appendix B.1.2`), enabling symbolic constraint induction from unstructed natural language feedback, making the system robust to real-world ambiguity.
>
> ---
>
> ### Writing issues:
>
>
> Thank you for carefully pointing out the writing issues. We appreciate your attention to detail and will carefully proofread the full manuscript to ensure clarity and correctness in the final version.
>
> ---
>
> We sincerely appreciate the thoughtful and concrete suggestions you raised. Your comments have prompted us to examine several important aspects of the method more carefully, and we believe they have meaningfully strengthened the revised manuscript.
>
> ---

---

> > ### Comment · Reviewer_fD4v · 2025-08-05
> >
> > The authors have provided detailed data and additional experiments that help address my concerns. While the new experiments are limited to comparisons with ProC3S, and many of the clarifications are framed primarily in relation to that work, the response nonetheless improves the overall clarity and reduces the logical gaps previously identified. I would adjust my score accordingly.

---

> > > ### Author Response · Authors · 2025-08-06
> > > **Thank you！**
> > >
> > > We sincerely thank you for your time and effort in reviewing our paper, and for your insightful comments that helped us better identify and clarify key aspects of our work. We're glad the revisions addressed your concerns, and appreciate your support in updating the score. Wishing you all the best.

---

### Official Review · Reviewer_MeGT · 2025-07-03

**Clarity:** 3
**Significance:** 2
**Originality:** 3
**Rating:** 4
**Confidence:** 4

**Summary:**

This paper presents InstructFlow, an LLM-based system that combines symbolic planning and constraint-guided code repair for long-horizon robotic manipulation. The central idea is to represent tasks via a hierarchical instruction graph, which is refined using symbolic feedback when execution failures occur. The system shows improved success rates on simulation benchmarks compared to PRoC3S, CaP, and LLM^3.

**Questions:**

What is the concrete benefit of using an instruction graph over simply sequencing high-level actions with reactive refinement?
Can you provide compelling examples where the graph becomes non-linear (e.g., has branches, merges, loops), and this structure is essential for solving the task?
Would a flat plan with dynamic constraint-aware sampling perform similarly, especially if paired with a constraint checker?
If you are using the same test suites in PRoC3S, why are the reported numbers different from the original paper? If the variance of the performance is high, maybe you need more trials?
Why is the generated constraint ``reusable’’? (line 331) The constraint is specific to a particular task setting/block positioning, why would it be reusable?
How would the proposed system generalize this to real-world tasks that cannot provide systematic feedback in textual format?

**Ethical Concerns:**

["NO or VERY MINOR ethics concerns only"]

**Final Justification:**

The proposed method is technically sound and demonstrates better performance than the SoTA prior work on benchmark tasks. However, the type of feedback required is unrealistic for real applications on physical platforms.

**Limitations:**

yes

**Quality:**

3

**Strengths And Weaknesses:**

Strengths:
The separation between planner, code generator, and constraint agent is conceptually clean and enables modular plan repair.
By using declarative constraints and spatial relations, the method enables interpretable and targeted adjustments.
The system outperforms baselines across several task domains and includes ablation studies to isolate component contributions.

Weaknesses:
The rationale behind structuring tasks as a symbolic instruction graph is not well-justified. It’s unclear why this representation is fundamentally superior to flat sequence planning or dynamic reactive strategies. The benefit seems incremental.
The paper primarily demonstrates simple, linear subgoal sequences. There’s no evidence that the instruction graph needs to go beyond a list of ordered subtasks. Are there any realistic scenarios where the graph structure becomes nonlinear (e.g., with branching, loops, or dependencies)?
The method relies on accurate symbolic representations of the scene and object poses. It is unclear how it would generalize to perceptual uncertainty or sensor noise, common in real-world settings.
All experiments are in simulation. No attempt is made to show robustness in hardware or even perception-grounded simulation with visual inputs.
While symbolic constraint induction is novel, the system appears to rely on a hand-designed set of predicates, limiting generality and scalability.

---

> ### Author Rebuttal · Authors · 2025-07-31
>
> We sincerely thank you for your careful review and thoughtful understanding of our work. Below, we provide detailed responses to your comments and describe the corresponding revisions made.
>
> ### Weakness 1:
>
> > *“The **rationale behind structuring tasks as a symbolic instruction graph** is not well-justified. It’s unclear why this representation is **fundamentally superior to flat sequence planning** or dynamic reactive strategies. The benefit seems incremental.”*
>
> ### Question 1:
>
> > *“What is the concrete **benefit of using an instruction graph** over simply sequencing high-level actions with **reactive refinement**?”*
>
> **Response**:
>
> Unlike flat or reactive planners, **our symbolic instruction graph supports structured reasoning and localized repair, which are crucial for *long-horizon, constraint-sensitive* tasks.**
>
> 1. **Flat plans lack structural insight:** Flat/reactive methods re-sample or regenerate plans without understanding why failures occur. InstructFlow induces symbolic constraints and injects reasoning nodes (e.g., spatial analysis, object selection) to refine only the affected subgoals, enabling efficient, interpretable correction. **As formalized in `Eq. (3)`,** $\text{instr}^{(t)} = \text{Encode}(\{ v^{\text{reason}}\_{\mathcal{T}\_j} \}_{j=1}^{|\mathcal{V}\_{\text{reason}}^{(t)}|},\ v\_{\text{plan}}^{(t)})$, and $\text{code}^{(t)} = \text{LLM}(\text{instr}^{(t)})$ , each code generation prompt is composed by integrating the outputs of upstream reasoning nodes in the instruction graph. This structured composition enables rich, context-aware prompting that flat sequences cannot support, leading to more grounded and accurate code synthesis.
>
> 2. **The benefit is significant:** As shown in `Table 2 (lines 299–300)`, removing the instructflow planner leads to 30–50% drops in complex tasks like Pyramid and Packing, indicating fundamentally different planning behavior.
>
> ---
>
> ### Weakness 2:
>
> > *“The paper primarily demonstrates simple, linear subgoal sequences. There’s no evidence that the instruction graph needs to go beyond a list of ordered subtasks.”*
>
>
> ### Question 2:
>
> > *“Can you provide compelling examples where the **graph becomes non-linear** (e.g., has branches, merges, loops), and this structure is essential for solving the task?”*
>
> **Response**:
>
> We agree that the final execution plan is often sequential, but the **instruction graph is not an execution trace, it is a symbolic reasoning graph that encodes how subgoals are derived, refined, and repaired.**
>
> Reasoning nodes often branch to multiple subgoals. When failures occur, we inject new reasoning nodes that modify only the affected subgraph, enabling structural repair, which flat lists cannot express.
>
> ---
>
> ### Weakness 3:
>
> > *“It is unclear **how it would generalize to perceptual uncertainty or sensor noise**, common in real-world settings.”*
> >
>
> **Response**:
>
> To address this concern, we added a controlled experiment simulating **perceptual uncertainty**. We **inject Gaussian perturbations into object poses** before symbolic abstraction. We test InstructFlow on Arrange and Arrange-YCB tasks across noise levels $σ\in\{0.005, 0.01, 0.02\}$, given that the average block_size is 0.04 (i.e., σ = 0.02 simulates severe, 50% object size noise). This setting reflects real-world errors and allows us to investigate robustness under increasing uncertainty.
>
> |  Table 1 (Reviewer-MeGT) | Pyramid | Line | Packing | Unstack | YCB-Packing | YCB-Stacking | Averge Drop    |
> | --------------------- | ------- |:----:| ------- | ------- | ------- | -------- | --- |
> | InstructFlow          | 90%     | 100% | 90%     | 90%     | 60%     | 70%      |   -  |
> | InstructFlow w/0.005  | 90%     | 100% | 90%     | 90%     | 60&     | 70%      |   0%  |
> | InstructFlow w/0.01   | 80%     | 100% | 80%     | 80%     | 50%     | 60%      |  8.33%   |
> | InstructFlow w/0.02   | 70%     | 90%  | 70%     | 70%     | 40%     | 50%      |  18.33%   |
>
> ---
>
>
> ### Weakness 4:
>
> > *“No attempt is made to show **robustness in hardware** or even **perception-grounded simulation with visual inputs**.”*
> >
>
> **Response**:
>
> In robotic manipulation, perception primarily serves to estimate object-level states (e.g., poses, identities), which downstream planners operate on. Our simulation provides these directly, allowing us to focus on planning.
>
> That said, we include a perception-grounded experiment in **`Appendix B.2.1 (Lines 574-575, Table 3)`**, where symbolic inputs are derived from visual observations via a VLM. This confirms that InstructFlow can operate with visual grounding and improves performance.
>
> Further, as noted in response to Weakness 3, we **simulate noisy perception** via pose perturbations (Table 1), and InstructFlow remains robust, supporting its potential for real-world deployment.
>
> ---
>
> ### Weakness 5:
>
> > *“While symbolic constraint induction is novel, the system appears to **rely on a hand-designed set of predicates**, limiting generality and scalability.”*
> >
>
> **Response**:
>
> InstructFlow does not rely on a fixed or hand-designed predicate set. **Instead, symbolic constraints are dynamically induced from execution feedback via spatial analysis, physical diagnostics, and semantic reasoning.**
>
> As detailed in **`Section 3.2`**, the Constraint Generator constructs symbolic predicates directly from failure contex, without referencing a predefined library. This process is formalized in Equation (4): $\mathcal{C}(\mathcal{E}, \mathcal{R}, \mathcal{F}, \mathcal{B}) = \{ R_i(e_{a_i}, e_{b_i}) \} \cup \{ f_j(\Theta_j) \oplus \tau_j \}$
>
> Here, the constraint set $\mathcal{C}$ is constructed from:
>
> 1. **Relational constraints** $R_i(e_{a_i}, e_{b_i})$, which capture spatial and logical relations between entities ;
> 2. **Physical constraints** $f_j(\Theta_j) \oplus \tau_j$, which encode quantitative properties like distance, stability, or alignment thresholds.
>
> **`Appendix B.2.2`** `(Lines 611-612, Table 4)` further illustrates  diverse predicates automatically generated across tasks.
>
> ---
>
> ### Question 3:
>
> > *“Would a **flat plan with dynamic constraint-aware sampling** perform similarly, especially if paired with a constraint checker?”*
> >
>
> **Response**:
>
> While flat plans with constraint-aware sampling (e.g., PRoC3S) improve robustness over naïve outputs, **they lack the ability to explain failures or perform targeted plan repair.** InstructFlow introduces two key enhancements:
> 1. **Symbolic diagnosis:** Failures trigger constraint induction (e.g., `ClearOf, Aligned`) to identify root causes.
> 2. **Graph-guided repair:** Only affected subgoals are updated via reasoning nodes, enabling localized and interpretable refinement.
>
> As shown in `abaltion study(lines 299–300, Table 2)`, removing either module leads to **30–50%** drops in success across complex tasks, confirming that symbolic abstraction adds essential value beyond flat planning.
>
> ---
>
> ### Question 4:
>
>
> > *“If you are using the same test suites in PRoC3S, why are the **reported numbers different from the original paper**?”*
> >
>
> **Response**:
>
> While we use the same environments and codebase as PRoC3S, its **original evaluation protocol** considers an execution successful if no simulator-level constraint is violated, **even when the task goal is not actually achieved** (e.g., misaligned stacks or wrong object placements).
>
> This issue is documented in `Appendix B.1.2 Fixing Protocol Limitations in PRoC3S Evaluation`, where we clearly provide example cases when PRoC3S consideres success (pass constraints) but actually fail (task goal) in `Figure 6 (Lines 560-561)`. **Same issue has been discussed in prior papers such as AHA [2]**.
>
> To address this gap, **we introduce a VLM-based semantic check that verifies whether the final scene satisfies the natural language goal.** `Figure 7 (Lines 565-566)` showcase our VLM-based success detector implementation. This evaluation protocol applies uniformly to all methods.
>
> As a result, **our reported success rates may be lower than PRoC3S's original numbers, not due to task changes, but due to stricter, goal-consistent evaluation for fairer comparison.**
>
> [1] Trust the proc3s: Solving long-horizon robotics problems with llms and constraint satisfaction. (CoRL'2024).
>
> [2] AHA: A Vision-Language-Model for Detecting and Reasoning Over Failures in Robotic Manipulation (ICLR'2025).
>
>
> ---
>
> ### Question 5:
>
> > *“Why is the **generated constraint reusable**? The constraint is specific to a particular task setting/block positioning.”*
>
> **Response**:
>
> By “reusable,” we refer to the **symbolic form** of constraints (e.g., `ClearOf(gripper, ?object`), `Below(?a, ?b)`), which capture **task-agnostic physical relations** rather than scene-specific instances.
>
> As shown in `Appendix Table 4`, predicates like `ClearOf`, `PlacementFeasible`, and `Aligned` recur across tasks such as Unstack, Packing, Stacking, and Pyramid, reflecting shared structural demands like collision avoidance, stable placement, and alignment.
>
> ---
>
>
> ### Question 6:
>
> > *“How would the proposed system generalize this to **real-world tasks that cannot provide systematic feedback in textual** format?”*
> >
>
> **Response**:
>
> Our method does not rely on text format per se, but on identifying task-level violations through perceptual or geometric cues. The Constraint Generator uses available feedback to locate relevant code segments, then performs semantic reasoning over object states and task goals to infer symbolic constraints.
>
> As described in  `Appendix B.1.2 `, our evaluation already incorporates VLM-based semantic verification (GPT-4o) that checks goal satisfaction from images. This demonstrates that our system **can operate on unstructed perceptual feedback**.
>
> ---
>
> Thank you again for your detailed and constructive feedback. We greatly appreciate the opportunity to address these points and improve the clarity and robustness of our work.
>
> ---

---

> > ### Comment · Reviewer_MeGT · 2025-08-06
> > **thanks for the response**
> >
> > Thank the authors for the detailed responses to my questions. I am happy to raise my rating to 4 but the major limitation is still that the system relies on text-based feedback, the case study of using VLM to provide feedback does not provide quantitative evaluation. At the same time, VLMs are not capable of performing real-time collision detection as shown in the example figure.

---

> > > ### Author Response · Authors · 2025-08-06
> > > **Thank you!**
> > >
> > > We sincerely thank the reviewer for the thoughtful comments, detailed questions, and constructive follow-up. We’re truly grateful that our responses helped clarify the core contributions of the work.
> > >
> > > We appreciate your insightful concerns and would like to offer some clarification regarding our design motivation. InstructFlow is fundamentally a multi-LLM system, which is designed to be modality-agnostic. While we currently use text as a convenient interface, the core mechanism relies on the LLM’s ability to interpret failure signals, regardless of the input form. As the reviewer rightly pointed out, extending constraint induction to other modalities, e.g., vision inputs, is a promising direction, and we are actively exploring this path in future work.
> > >
> > > We take these concerns seriously and will clearly articulate both the limitations and the path forward in the revised version. Thank you again for your engagement and support, it genuinely motivates us to further improve the work. We wish you all the best in your work and life.

---

### Official Review · Reviewer_xtF7 · 2025-07-05

**Clarity:** 3
**Significance:** 3
**Originality:** 3
**Rating:** 5
**Confidence:** 4

**Summary:**

The paper introduces InstructFlow, a modular, multi-agent framework for robotic code generation. It uses a hierarchical instruction graph and symbolic constraint reasoning for task planning. The model takes advantage of dynamically feedback from execution failures, which it integrates into symbolic constraints.

**Questions:**

Questions:

- Would InstructFlow generalize across robotic domains beyond tabletop manipulation, e.g., mobile navigation, assembly?

- How much doe sthe method depend on the underlying language model (e.g., GPT-4 vs. smaller models)?

- Can the symbolic constraint vocabulary be learned or adapted online instead of being manually defined?

**Ethical Concerns:**

["NO or VERY MINOR ethics concerns only"]

**Final Justification:**

I have read the author response. I would suggest that they add the additional results and clarification in the final version, if accepted. I am raising my score, as the authors have done a lot of extra work.

**Limitations:**

No dedicated section on this.

**Quality:**

3

**Strengths And Weaknesses:**

Strengths:

- Combines probabilitisc and symbolic comnstraints for task planning, which I believe is novel.

- Sizable improvements across multiple domains compared to strong baselines like PRoC3S and LLM3: 20–40%

- Careful ablation analysis

- Interpretability of the produced plans

- Allows recovery from bad initial plans

- Detailed description of the experimental setup, including task domains, constraints, baselines, and compute resources, which should enable reproducisbility.


Weaknesses:

- All results are based on simulations; however, nowadays in robotics, it is standard to have also real-world experiments with actual robots.

- Limited kinds of tasks: table top manipulation

- No Limitations section (even though some limitations are discussed in the text).

Minor:

- The authors refer to intuition in a couple of places; this is not a scientific method.

---

> ### Author Rebuttal · Authors · 2025-07-31
>
> We are truly grateful for your generous evaluation and insightful feedback. We have reviewed your feedback thoroughly and respond below with clarifications and revisions made accordingly.
>
> ### Weakness 1:
>
> > “All results are based on simulations; however, nowadays in robotics, it is standard to have also **real-world experiments** with actual robots.”
>
>
> **Response**:
>
> We agree that real-robot validation is valuable. While our study is simulation-based, we explicitly model two key real-world uncertainties: sensor noise and feedback ambiguity, to evaluate robustness:
>
> 1. **Perceptual noise:** We inject Gaussian perturbations into object poses before symbolic abstraction to simulate sensor inaccuracies. As shown in **Table 1**, InstructFlow maintains strong performance even under severe noise (σ = 0.02, 50% of object size), with success rates remaining above 70% on most tasks.
>
>
> |    Table 1   (Reviewer-xtF7)   | Pyramid | Line | Packing | Unstack | YCB-Packing | YCB-Stacking | Averge Drop    |
> | --------------------- | ------- |:----:| ------- | ------- | ------- | -------- | --- |
> | InstructFlow          | 90%     | 100% | 90%     | 90%     | 60%     | 70%      |   -  |
> | InstructFlow $\sigma =$ 0.005  | 90%     | 100% | 90%     | 90%     | 60&     | 70%      |   0%  |
> | InstructFlow $\sigma =$ 0.01   | 80%     | 100% | 80%     | 80%     | 50%     | 60%      |  8.33%   |
> | InstructFlow $\sigma =$ 0.02   | 70%     | 90%  | 70%     | 70%     | 40%     | 50%      |  18.33%   |
>
> 2. **Feedback-layer noise:** We evaluate robustness under incorrect and incomplete failure traces, simulating real-world ambiguity.  (1) incorrect feedbacks with wrong object references, and (2) incomplete feedbacks with missing object IDs or causes. Across 10 tasks (Table 2 below), InstructFlow consistently achieves near-baseline performance despite missing or wrong feedback information.
>
> | Table 2   (Reviewer-xtF7)    | Star | Arrow | Letters | Enclosed | Pyramid | Line | Packing | Unstack | YCB-Packing | YCB-Stacking | Average Drop    |
> | -------------------------------- | ---- |:-----:| ------- | -------- | ------- | ---- | ------- | ------- | ------- | -------- | --- |
> | InstructFlow                     | 100% |  80%  | 100%    | 100%     | 90%     | 100% | 90%     | 90%     | 60%     | 70%      |  -   |
> | InstructFlow w/incorrect feedback  | 100% |  80%  | 100%    | 100%     | 60%     | 80%  | 70%     | 60%     | 30%     | 40%      |   16%  |
> | InstructFlow w/incomplete feedback | 100% |  80%  | 100%    | 100%     | 70%     | 90%  | 70%     | 70%     | 40%     | 50%      |   11%  |
>
> These results show InstructFlow operates reliably without perfect state or supervision，a key requirement for real-world deployment. We view hardware validation as important future work and believe these robustness results provide strong evidence of real-world readiness.
>
> ---
>
> ### Weakness 2:
>
> > “Limited kinds of tasks: **table top manipulation.** ”
>
> ### Question 1:
>
> > “Would InstructFlow **generalize across robotic domains beyond tabletop manipulation**, e.g., mobile navigation, assembly?”
>
> **Response**:
>
> We appreciate the reviewer’s question on generality. While our experiments focus on tabletop manipulation for benchmarking consistency (e.g., PRoC3S), **InstructFlow is domain-agnostic by design**.
>
> InstructFlow operates over **textual instructions** and **symbolic predicates**, enabling generalization beyond tabletop setups. Its key components, symbolic instruction graph planning, modular node composition, and failure-driven constraint repair, reason over abstract subgoals without relying on domain-specific assumptions (e.g., fixed workplace or rigid objects).
>
> 1. In **mobile navigation**, pfailures like deviation or blocked paths naturally yield predicates such as `PathBlocked` or `DeviationTooLarge`, enabling symbolic route repair.
> 2. In **assembly domains**, misalignment or insertion failure can trigger constraints like `AlignedForInsertion` or `GripTorqueInsufficient`, driving targeted graph refinement.
>
> These examples illustrate how InstructFlow’s **planning-repair cycle generalizes to other robotic domains**, and we view broader deployment (e.g., mobile or deformable tasks) as promising future work.
>
> ---
>
> ### Weakness 3:
>
> > “No **Limitations section** (even though some limitations are discussed in the text).”
> >
>
> **Response**:
>
> Thank you for pointing this out. While we discussed key limitations throughout the paper, we agree that a dedicated section would improve clarity.
> - As noted in **`Lines 365–367`**, InstructFlow currently assumes structured symbolic inputs and limited sensory grounding. We are actively exploring multimodal extensions with visual grounding and constraint induction from unstructured feedback.
> - As discussed in **`Appendix B.2.3`**, performance degrades in scenarios with **irregular object geometries** and **tight spatial coupling** (e.g., Packing-YCB, Stacking-YCB), due to the increased difficulty of reliable constraint induction.
>
> We will add a dedicated Limitations section in the final version to consolidate these observations.
>
> ---
>
> ### Question 2:
>
> “How much does the method depend on the underlying language model (e.g., **GPT-4 vs. smaller models**)?”
>
>
> **Response**:
>
> Thank you for raising this point. We conducted additional ablation experiments usign **GPT-3.5-turbo** in place of GPT-4o across all and individual modules.
>
>
> | Table 3   (Reviewer-xtF7)    | Star | Arrow | Letters | Enclosed | Pyramid | Line | Packing | Unstack | YCB-Packing | YCB-Stacking |  Average Drop   |
> | ---------------------------------------- | ---- |:-----:| ------- | -------- | ------- | ---- | ------- | ------- | ------- | -------- | --- |
> | InstructFlow w/**GPT-4o**                | 100% |  80%  | 100%    | 100%     | 90%     | 100% | 90%     | 90%     | 60%     | 70%      | -  |
> | InstructFlow w/**GPT-3.5-turbo**         | 90%  |  70%  | 90%     | 100%     | 60%     | 80%  | 70%     | 60%     | 40%     | 50%      |   17%    |
> | Code generator w/**GPT-3.5-turbo**       | 90%  |  70%  | 100%    | 100%     | 70%     | 80%  | 70%     | 60%     | 50%     | 60%      |    13%   |
> | Constraint generator w/**GPT-3.5-turbo** | 90% |  80%  | 100%    | 100%     | 70%     | 90% | 90%     | 70%     | 50%     | 60%      |   8%  |     |
> | Planner w/**GPT-3.5-turbo**              | 90% |  80%  | 100%    | 100%     | 80%     | 100% | 90%     | 80%     | 60%     | 60%      |   4%    |
>
> 1. **Code generator is model-sensitive**: Substituting GPT-4o with GPT-3.5 in this module results in the largest performance drop (~**13%**), highlighting its reliance on strong LLM reasoning for reliable code synthesis, especially in long-horizon, constraint-sensitive tasks.
> 2. **Planner and Constraint Generator are robust:** These modules maintain competitive performance with GPT-3.5 (**4–8% drop**), demonstrating that symbolic abstraction and structured reasoning mitigate dependence on large-scale models.
>
> These findings suggest that InstructFlow’s **core effectiveness stems from its agentic design**, not LLM size. It remains effective with smaller models in key components, making it practical for resource-limited deployments.
>
>
> ---
>
> ### Question 3:
>
> > “Can the **symbolic constraint vocabulary be learned or adapted online** instead of being manually defined?”
> >
>
> **Response**:
>
> The constraint vocabulary in InstructFlow is **not manually defined nor fixed**. Instead, **symbolic predicates are dynamically induced online** from failure traces using spatial analysis, physical diagnostics, and semantic reasoning (**`Section 3.2`**).
>
> Formally (`Lines 266`, **`Eq. 4`**), $\mathcal{C}(\mathcal{E}, \mathcal{R}, \mathcal{F}, \mathcal{B}) = \{ R_i(e_{a_i}, e_{b_i}) \} \cup \{ f_j(\Theta_j) \oplus \tau_j \}$, the Constraint Generator constructs task-specific constraints $\mathcal{C}$ from:
> 1. **Relational constraints** $R_i(e_{a_i}, e_{b_i})$ over entity pairs (e.g., reachability, proximity)
> 2. **Physical constraints** $f_j(\Theta_j) \oplus \tau_j$ capturing measurable task variables (e.g., alignment thresholds, support stability).
>
> These predicates are constructed per execution context without reference to any pre-defined library. As illustrated in `Appendix B.2.2 (Lines 611-612, Table 4)`, InstructFlow generates diverse symbolic constraints across tasks, supporting adaptability and generalization.
>
> We view learning a more generalized or transferable constraint space across tasks as an exciting direction for future work.
>
> ---
>
> We sincerely thank you for your encouraging evaluation and thoughtful comments. Your recognition of our work’s contributions is highly motivating, and your suggestions have helped us further strengthen the clarity and rigor of the manuscript.
>
> ---

---

> > ### Comment · Reviewer_xtF7 · 2025-08-06
> >
> > I have read the author response. I would suggest that they add the additional results and clarification in the final version, if accepted. I am raising my score, as the authors have done a lot of extra work.

---

> > > ### Author Response · Authors · 2025-08-06
> > > **Thank you!**
> > >
> > > Thank you for your thoughtful review and encouraging response. We will make sure to include the new results and clarifications in the final version, and truly appreciate your recognition of our additional efforts. If you have any further suggestions, it would be our pleasure to hear them.  Best wishes for your future work.

---

### Decision · Program_Chairs · 2025-09-17

**Decision:**

Accept (poster)

**Comment:**

The paper introduces InstructFlow, a framework for robotic code generation that addresses failures in long-horizon planning tasks. It structures task prompts into a hierarchical instruction graph of subgoals, which are translated into executable code by a code generator, while a constraint generator diagnoses execution failures and induces symbolic constraints for targeted plan repair. Unlike prior flat or reactive methods, InstructFlow continuously integrates feedback into symbolic reasoning, enabling interpretable and adaptive code refinement without regenerating full plans. Experiments in drawing, block stacking, YCB object packing, and other tasks show that InstructFlow improves success rates by 20–40% over strong LLM-based baselines.

One of the issues with this work is that while symbolic constraints are induced automatically, the framework still depends on a curated set of predicates, diagnostic rules, and failure categories, which may limit scalability across unseen tasks or domains. Additionally, the system assumes structured state information (object poses, relations), and does not yet incorporate raw visual perception or noisy sensor data.

Given recent progress in reasoning for robotic task planning, I don't necessarily see this paper as significantly standing out in terms of novelty compared to methods such as ProC3S, or RoCo: Dialectic Multi-Robot Collaboration with Large Language Models, but it is a useful addition to the literature.

Many of the concerns of reviewers were addressed during the rebuttal and despite the issues mentioned above, there seems to be consensus that this paper should be accepted, which is what I am inclined to recommend as well.